# Technical note: Surface fields for global environmental modelling

Margarita Choulga[1], Francesca Moschini[1], Cinzia Mazzetti[1], Stefania Grimaldi[2], Juliana Disperati[3], Hylke Beck[4], Peter Salamon[2], Christel Prudhomme[1]

[1]European Centre for Medium-Range Weather Forecasts (ECMWF), Reading, RG2 9AX, United Kingdom
[2]Joint Research Centre (JRC), European Commission, Ispra, 21027, Italy
[3]Fincons Group, Vimercate, 20871, Italy
[4]King Abdullah University of Science and Technology (KAUST), Thuwal, Saudi Arabia

*Correspondence to*: Margarita Choulga (margarita.choulga@ecmwf.int) and Christel Prudhomme (christel.prudhomme@ecmwf.int)

**Abstract.** Climate change has resulted in more frequent occurrences of extreme events, such as flooding and heavy snowfall, which can have a significant impact on densely populated or industrialised areas. Numerical models are used to simulate and predict these extreme events, enabling informed decision-making and planning to minimise human casualties and protect costly infrastructure. LISFLOOD is an integrated hydrological model underpinning the European and Global Flood Awareness Systems (EFAS and GloFAS, respectively) developed by the Copernicus Emergency Management Service (CEMS). The CEMS_SurfaceFields_2022 dataset is a new set of high-resolution surface fields at 1 and 3 arc min (approximately 2 and 6 km at the equator respectively) covering Europe and the global land surface (excluding Antarctica) respectively, based on a wide variety of high-resolution and up-to-date data sources. The dataset encompasses (i) catchment morphology and river network, (ii) land use, (iii) vegetation cover type and properties, (iv) soil properties, (v) lake information, and (vi) water demand. This manuscript details the complete workflow to generate CEMS_SurfaceFields_2022 fields, including data sources and methodology. Whilst created together with upgrades to the open source LISFLOOD code, the CEMS_ SurfaceFields_2022 fields can be used independently for a wide range of applications, including as input of hydrological, Earth System or environmental modelling, or for carrying out general analyses across spatial scales, ranging from global and regional to local levels (especially useful for regions outside Europe), expected to improve accuracy, detail, and realism of applications.

## 1 Introduction

Current numerical Earth system models are highly complex. Thanks to the availability of High Performance Computers, cloud computing, and a wide range of high-resolution environmental data derived from the use of ground, unconventional and satellite measurement sensors, numerical global models are even able to reach kilometre-scale horizontal resolution. But increase in spatial resolution also means that the Earth system and environmental models have to represent more surface and atmospheric processes and their interactions, which can become challenging, for example in complex orographic areas. Model accuracy heavily depends on the quality of the input surface fields (i.e. how realistic and up-to-date they are), and it is essential to minimise errors in surface fields. New high-resolution (i.e. 10-100 m) surface datasets based on daily satellite observations are now frequently released and continuously supported by e.g. the Copernicus program (e.g. Global Land Cover: Buchhorn et al., 2021; GHSL-BUILT-S: Pesaresi and Politis, 2022; Schiavina et al., 2022), which helps in achieving the goal of minimising surface field errors. It was shown, e.g. in Kimpson et al. (2023), that the use of accurate and up-to-date underlying information to generate model's input surface fields can substantially reduce skin temperature errors even at 30 km horizontal resolution (Kimpson et al., 2023).

Following the digital revolution of cloud archiving and computing, where data, software and information technology (IT) infrastructure can be accessed by anyone from everywhere, the Earth systems and environmental modelling community has also moved from codes developed by a single organisation and few contributors, to so-called 'community models' where a reference code is open for free use and/ or development according to sharing principles. Such models include Joint UK Environmental Simulator JULES, a land surface model whose development is coordinated by the UK Met Office and UKCEH (Best et al., 2011; Clark et al., 2011; Marthews et al., 2022), OpenIFS, a Numerical Weather Forecast model available to external users for research and training (Sparrow et al., 2021; Carver, 2022; Huijnen et al., 2022; Köhler et al., 2023), the Community Land Model CLM, an Earth System Model with strong climate component maintained by the National Centre for Atmospheric Research but available for use by the wider research community (Lawrence et al., 2019), or LISFLOOD-OS, a spatially distributed water resources model developed by the Joint Research Centre (JRC; Van Der Knijff and De Roo, 2008) and available for use and development through a share code repository (available online: https://ec-jrc.github.io/lisflood/#lisflood; https://ec-jrc.github.io/lisflood-code/, last accessed: 21.01.2024).

| 53 | To promote the seamless development of science, and facilitate research community efforts in working with the |
| 54 | same code and input data, providing feedback, and improving the code and the data itself, powerful web-based |
| 55 | platforms can be used. One of them is the Google Earth Engine (GEE; Gorelick et al., 2017), a free-of-charge |
| 56 | platform that provides easy, web-based access to an extensive catalogue of satellite imagery and other geospatial |
| 57 | data in an analysis-ready format. The data catalogue is embedded into Google computing platform that lets you |
| 58 | easily implement all personal workflows, which facilitates global-scale analysis and visualization (GEE: FAQ, |
| 59 | 2023). GEE was chosen for the generation of a new vast surface field set due to its high resolution data catalogue |
| 60 | and powerful computation capabilities. |
| 61 | This manuscript presents the methodology used to prepare the CEMS_SurfaceFields_2022 dataset containing all |
| 62 | surface fields necessary to run the LISFLOOD-OS model at resolutions ~2 km at the equator or 1 arc min (over |
| 63 | Europe) and ~6 km at the equator or 3 arc min (globally). CEMS_SurfaceFields_2022 were used in the set-up of |
| 64 | the Early Warning Systems of the Copernicus Emergency Management Service of the European Union for the |
| 65 | European (European Flood Awareness System EFAS version 5; Smith et al., 2016; information available online: |
| 66 | https://www.efas.eu/, last accessed: 21.01.2024) and global (Global Flood Awareness System GloFAS version 4; |
| 67 | Hirpa et al., 2018; Alfieri et al., 2020; Harrigan et al., 2023; information available online: |
| 68 | https://www.globalfloods.eu/, last accessed: 21.01.2024) domains operational in December 2023 (EFASv5 and |
| 69 | GloFASv4). Details on raw data collection, scientific protocol, and technical methods aim to allow the adequate |
| 70 | understanding and interpretation of the surface field datasets, and for any interested user to generate their own |
| 71 | datasets by replicating or adapting the workflow to different fields, geographical domain, spatial resolution, or |
| 72 | content as relevant for downstream application. The manuscript is structured as follows: Section 2 provides an |
| 73 | overview of the surface fields, explains the criteria to select reference data, where and how they were processed, |
| 74 | and outlines the general methodology to produce the surface fields; Section 3 to Section 8 details the reference |
| 75 | data and specific methodology applied to each surface filed category, including examples of application; Section |
| 76 | 9 provides all the relevant information for data access; Section 10 discusses the challenges of creating a consistent |
| 77 | high resolution continental and global scale set of consistent surface fields and the opportunities disclosed by their |
| 78 | availability. |

## 2 Surface fields for distributed environmental modelling

### 2.1 General information

| 81 | Environmental models, especially land surface and hydrological models, simulate how water moves across |
| 82 | canopy, surface, subsurface, ground and eventually river channels using mechanistic equations that describe the |
| 83 | physics of these processes. Each model represents processes with more or less complexity, depending on the |
| 84 | model purpose and expected output (Rosbjerg and Madsen, 2006). With most represented terrestrial processes |
| 85 | depending on the landscape, information describing the spatial variation in the geophysical and vegetation |
| 86 | characteristics is needed. Such characteristics include morphological features (e.g. channel geometry, orography |
| 87 | or slope), soil hydraulic property, land and vegetation features (e.g. ecosystem cover type, leaf area index (LAI), |
| 88 | evaporation rates, crop type, planting and harvesting dates), and if relevant, human intervention information such |
| 89 | as population density or type of water usage. |
| 90 | LISFLOOD is a semi-distributed, physically based hydrological model which has been designed for the modelling |
| 91 | of rainfall-runoff processes in large and transnational catchments (Bates and De Roo, 2000; De Roo et al., 2000; |
| 92 | De Roo et al., 2001; Van Der Knijff and De Roo, 2008; Van Der Knijff et al., 2010; Burek et al., 2013). In its |
| 93 | most prominent application, LISFLOOD is used by the Copernicus Emergency Management Services' EFAS and |
| 94 | GloFAS to provide medium range and seasonal riverine flow forecasts (Alfieri et al., 2020). LISFLOOD is also |
| 95 | widely used for a variety of applications, including water resources assessment (drought forecast); analysis of the |
| 96 | impacts of land use changes, river regulation measures, water management plans; climate change analysis (e.g. |
| 97 | Vanham et al., 2021). |
| 98 | To facilitate users' uptake and enable the seamless development of science, LISFLOOD has been released as open |
| 99 | source in 2019, i.e. LISFLOOD-OS. The open-source suite includes the LISFLOOD hydrological model and a set |
| 100 | of auxiliary tools for model setup, calibration, and post-processing of the results. For instance, the pre-processor |
| 101 | LISFLOOD-LISVAP can be used to compute evapotranspiration, which is one of the three meteorological |
| 102 | variables, along with total precipitation and average temperature, strictly required as input to the hydrological |
| 103 | model. |
| 104 | The modelling of runoff processes in different climates and socio-economic contexts then requires a set of raster |
| 105 | fields (i.e. set of surface fields presented in this manuscript) to provide information of terrain morphology, surface |
| 106 | water bodies, soil properties, land cover and land use features, water demand. The total number of fields range |
| 107 | between 66, when only the essential rainfall-runoff processes are modelled, to a total 108 for a more |

comprehensive model set-up in which, for instance, lakes, reservoirs, water demand for anthropogenic use are
included (available online: https://ec-jrc.github.io/lisflood-model/, last accessed: 21.01.2024).
The main model's field (i.e. in technical for model operation/ running sense) is 'mask' – a Boolean field that
defines model boundaries, i.e. grid cells over which the model performs calculations and grid cells which are
skipped (e.g. ocean grid cells). Whilst the surface fields described in this manuscript follow specific requirements
of the LISFLOOD-OS model, they are a source of versatile information that can be used for any environmental
modelling application, either directly, or following a transformation, as relevant, as a full set or as a few consistent
fields.

## 2.2 Reference data and methodology

To produce CEMS_SurfaceFields_2022 surface fields only open source, freely available, updated as recently as
possible, with recognised reference on their quality data sources were used (see Appendix 1 for all relevant
reference data details). Note that whilst the majority of surface fields contain no time element, vegetation and
water demand fields explicitly describe the annual cycle (vegetation, rice) or annual time evolution (water
demand) and therefore have more stringent requirements regarding the data source. Global single-source datasets
(e.g. Te Chow, 1959; Supit et al., 1994; Allen et al., 1998; Buchhorn et al., 2021) were favoured to regional and/
or multiple data sources that needed to be combined in order to produce the required data unless sub-set
information was of much better quality (e.g. Moiret-Guigand, 2021). CEMS_SurfaceFields_2022 surface fields
are based on 25 different data sources and consist of 140 gridded fields grouped into six following groups: (i)
catchment morphology and river network, (ii) land use, (iii) vegetation cover type and properties, (iv) soil
properties, (v) lake information, and (vi) water demand.
Considering the high resolution (i.e. hundreds of meters) and volume of data (i.e. GB) of most input datasets used
to generate the surface fields, a high performing data manipulation platform was needed. GEE (Gorelick et al.,
2017) was selected as it provides (embedded) a vast high resolution data catalogue (e.g. ready available MERIT
DEM elevation dataset, CGLS-LC100 and CLC2018 land cover datasets) and powerful computation capabilities.
It also allows to upload any raster and vector data (e.g. GeoTiff or shapefiles) and to conduct each surface field
tailored computations. All GEE scripts were written in JavaScript to produce GeoTiff files, converted to the final
file format (NetCDF) locally after transfer from GEE platform.
To ensure a consistent representation of physical processes at all scales, surface fields should be as coherent as
possible among each other – between variables and across scales. Coherency can be achieved by using, where
possible, the same input datasets to derive different field types (e.g. unique forest information input to create all
forest-related surface fields), and making sure spatial aggregation or disaggregation across scales results in
expected values. Figure 1 shows a simplified scheme that relates input datasets (e.g. CGLS-LC100) with the
resulting surface fields (e.g. surface cover fractions – forest, inland water, and sealed surface fraction fields), also
highlighting fields requiring intermediary and sequential steps (e.g. forest fraction is needed to create soil
parameter fields over forested and non-forested areas).
For processes with horizontal dependency such as river routing, the relationship between grid cells (e.g. how the
grid cells are connected) must be defined first so that all dependent fields can be generated on the same grid
coordinates, spatial resolution and using consistent input data. For example, LDD defines how water moves across
the model grid cells as a river drainage network (see Figure 2) and strongly depends on elevation data (see Section
3 for more details). Because of the complex spatial dependency of a river drainage network, LDD must be created
directly from elevation data at the required grid and resolution and cannot be resampled from a previous LDD
field of a different grid and/ or resolution. It is then used to define information on the river network, including
upstream drainage area and gradient. Note, Figure 1 misses an arrow from MERIT DEM to LDD only because
this step was mainly done by CaMa-Flood developers (see Section 3.2 for more details).
Four steps are involved in generating a particular surface field (see Table 1), with step 3 being the most complex
and varied (see Figure 2 for an example), and step 4 being necessary only for some model specifications (here as
required by LISFLOOD, see Table 2).
All techniques applied (see Table 1) to generate CEMS_SurfaceFields_2022 are reproduceable to different input
data and/ or for different output data specifications. Further details on specific manipulations associated with each
field category are given in sections below as relevant, where each section has a table with exact data source used
per surface field, and step-by-step description of transformations applied to the data to compute the final fields
included in CEMS_SurfaceFields_2022 (full technical descriptions for all fields are explained in the LISFLOOD
user guide, available online: https://ec-jrc.github.io/lisflood-code/4_Static-Maps-introduction/, last accessed:
21.01.2024). Although the specific requirements for the dataset were defined by LISFLOOD for EFAS (European
domain, 1 arc min resolution at mid-latitude of the domain (47.50 N) is ~1.25 km) and GloFAS (Global domain)
implementation, summarised in Table 2, they are consistent with requirements of any other environmental models.
Regional examples of a sub-set of CEMS_SurfaceFields_2022 are provided to show the level of detail available
at each resolution and field, and to emphasise the consistency through all the fields, a critical requirement for
environment modelling and analysis, focusing on three regions of the world: the Po River (Europe), the Amazon
River (South America) and the Brahmaputra River (Asia), with additional examples provided in Appendix 4).

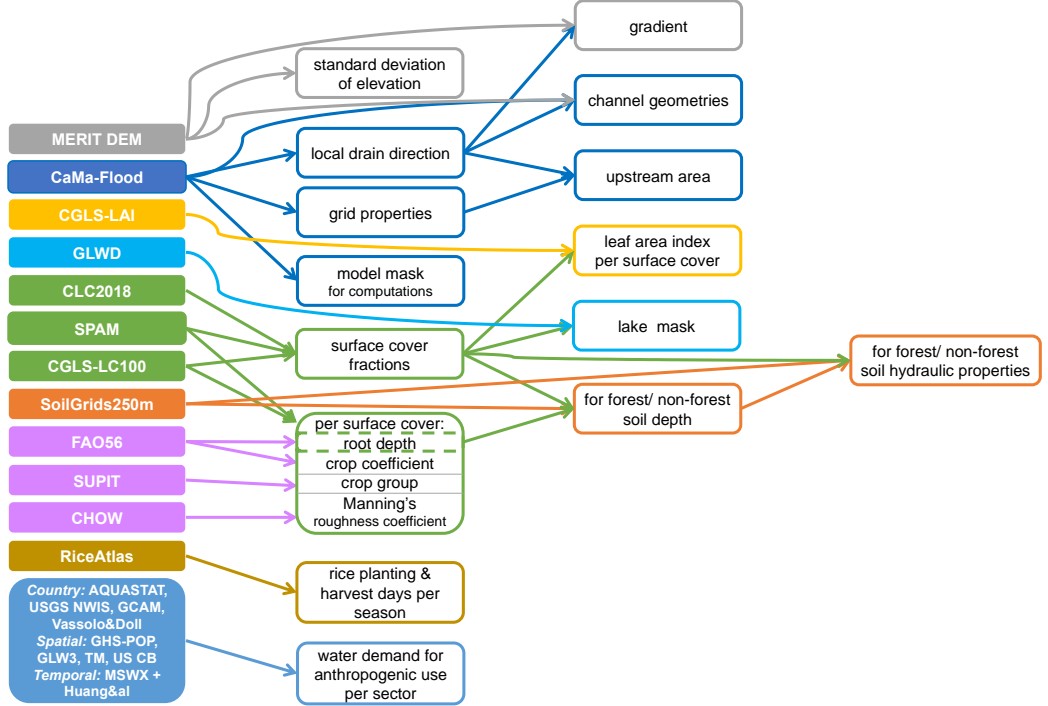

**Figure 1. Flow chart connecting input datasets and surface fields created. Dashed border denotes intermediate fields,**
**that are not part of the final dataset catalogue.**

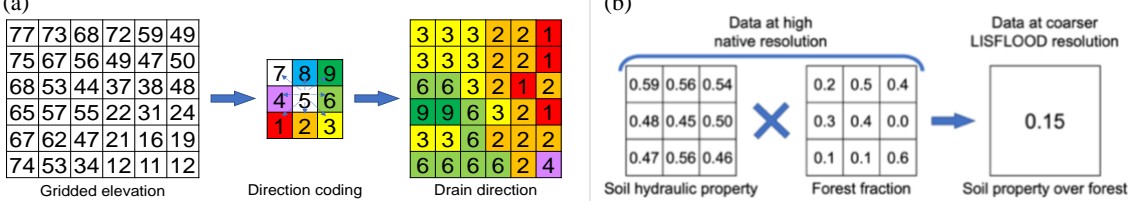

**Figure 2. Examples of data manipulation for (left column, plot a) transformation of elevation data into LDD (done**
**within CaMa-Flood), and (right column, plot b) upscaling with weighted average for one final grid cell of soil hydraulic**
**property over forested area.**
**Table 1. The four steps of a particular surface field generation and associated data manipulations.**

| Order | Description | Purpose | Function |
|---|---|---|---|
| 1 | Raw file preparation | Vector gridding, region merging | |
| | | Upscaling (spatial/ temporal aggregation) | Arithmetic mean, mode, sum, standard deviation (weighted) resampling from auxiliary data |
| 2 | Unit conversion | Converting values from native to fraction per grid cell | Surface area, percentage or categorical to fractions per grid cell (see Appendix 2 for more details) |
| 3 | Value computation | Transforming | Mathematical equation/ function needed to generate the output variable |
| | | Reprojecting | Interpolation (changing grid, preserving resolution in meters) |
| | | Upscaling (spatial [default]/ temporal aggregation) | Arithmetic mean, mode, sum, standard deviation (weighted) resampling from auxiliary data (changing resolution, preserving grid) |
| | | Downscaling (spatial [default]/ temporal disaggregation) | Nearest neighbour (changing resolution, preserving grid) |
| | | Limiting | Force a minimum/ maximum value to satisfy e.g. calculation precision, physical meaning and/ or model requirement |
| 4 | Zero/ NoData filling | Replace zero/ NoData by the most appropriate values | LIGHT. Constant value, unweighted global mean, unweighted global mode |
| | | | DEEP. Values from next coarser resolution (up to an agreed maximum resolution); if still missing, method LIGHT |

**Table 2. Dataset files technical specifications.**

| *Type* | *Specification* |
|---|---|
| Format | NetCDF |
| Projection | EPSG:4326 - WGS84: World Geodetic System |
| Horizontal resolution | Europe: 1 arc min (~1.86 km at the equator) [file size 4530x2970 grid cells] |
| | Globe: 3 arc min (~5.57 km at the equator) [file size 7200x3600 grid cells] |
| Domain bound | Europe: [North = 72.25 N; South = 22.75 N; West = 25.25 W; East = 50.25 E] |
| | Globe: [North = 90.00 N; South = 90.00 S; West = 180.00 W; East = 180.00 E] |
| Missing value (i.e. NoData) location | Over land: none |
| | Over ocean: all ocean grid cells have missing value (i.e. ocean is masked based on 'mask' field) |
| Missing value (i.e. NoData) number | For Integer variable type: 0 |
| | For Real variable type: -999999.0 |
| Variable type | Integer: Int8 |
| | Real: Float32 |

**3 Catchment morphology and river network**
**3.1 General information**
Morphology and channel shape information are essential for the computation of snow melting, temperature
scaling, and river routing. Statistics such as standard deviation of elevation and other orographic sub-grid
parameters critical for radiation parametrization, especially for shadowing effect, whilst channel geometry fields
are needed to describe overbank inundation and infer inundated areas in wetland methane and soil carbon
modelling, for example. Land morphology is derived from elevation and its variability within a single cell can be
represented through slope, standard deviation, aspect, etc. River drainage information, derived from elevation, is
used to connect the model cells according to the direction of the surface runoff, with channel geometry information
used for routing processes.
The dataset contains 14 morphology and river network variables (name in brackets in italics correspond to the
field's name in the data repository):
• Morphologic information: local drainage direction (i.e. flow direction from one cell to another; *LDD*,
dimensionless), upstream drainage area (*upArea*, m$^2$), grid cell area (*pixarea*, m$^2$), grid cell length
(*pixleng*, m), standard deviation of elevation (*elvstd*, m), gradient (i.e. elevation gradient; *gradient*, m/m);
• Kinematic wave equation for routing: channel bottom width (*chanbw*, m), channel length (*chanlenght*,
m), channel gradient (*changrad*, m/m), Manning's roughness coefficient for channels (*chanman*, s/m$^{1/3}$);
• River network information: channel mask (i.e. presence of river channel; *chan*, dimensionless), channel
side slope (i.e. channel's horizontal distance divided by vertical distance; *chans*, m/m);
• Open water evaporation: bankfull channel depth (*chanbnkf*, m), channel flood plain (i.e. width of the area
where the surplus of water is distributed when the water level in the channel exceed the channel depth;
*chanflpn*, m).
**3.2 Reference data and methodology**
Environmental models require an accurate description of terrain and hydro-morphology to represent the
hydrodynamics at the spatial resolution of the model. Here all catchment morphology and river network fields are
derived from (i) **The Catchment-based Macro-scale Floodplain (CaMa-Flood) Global River Hydrodynamics**
**Model v4.0 maps** (further referred as CaMa-Flood) – that include information on channel length, river topography
parameters, floodplain elevation profile, channel width and channel depth at 3 and 1 arc min resolutions covering
land area from 90 N to 60 S, representative of the year 2017, and (ii) **The MERIT DEM: Multi-Error-Removed**
**Improved-Terrain Digital Elevation Model v.1.0.3** (further referred as MERIT DEM) – a high accuracy global
DEM at 3 arc second resolution (~90 m at the equator) covering land area from 90 N to 60 S, representative of
the year 2018 (for reference data details see Appendix 1). All fields follow a complex sequential workflow (see
Figure 3 and Table 1). Note that whilst some river network fields were already directly available from the CaMa-
Flood catalogue (e.g. LDD, channel length), they had to be adapted to the specific requirements of LISFLOOD,
specifically consistent with an interconnected river network described by the D8 algorithm (O'Callaghan and
Mark, 1984; Figure 2a) different to that used by the CaMa-Flood algorithm.

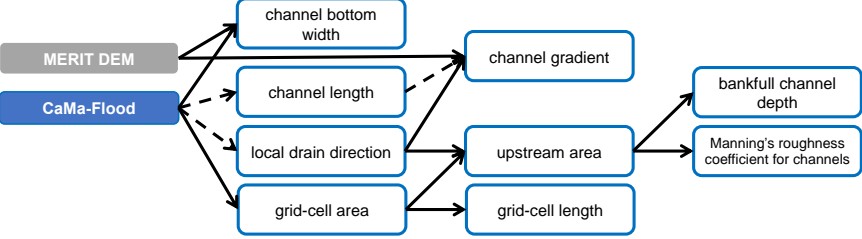

**Figure 3. Workflow of complex manipulations to create some of the morphology and river network fields; solid arrows**
**indicate a function transformation, dashed – modification of existing input data to LISFLOOD specifications.**
**Table 1. Morphology and river network fields, their description, data source and applied transformation; * denotes**
**transformation following Burek et al. (2014); name in brackets in italics next to each field correspond to the name in**
**the data repository.**

| Field type | Description | Data source (variable) | Transformation |
|---|---|---|---|
| Local drainage direction (*LDD*) | Connects every grid cell forming a river network from springs to mouth | CaMa-Flood (flwd) | Direction coding, ensuring grid cell connectivity |
| Grid cell area (*pixarea*) | Area of every grid cell | CaMa-Flood (flwd) | Grid cell area based on a given coordinate reference system and resolution |
| Grid cell length (*pixlength*) | Length of every grid cell | *pixarea* | $pixlength = \frac{pixarea}{resolution}$, where *resolution* – 1.86 km and 5.57 km for 1 and 3 arc min respectively |
| Upstream drainage area (*upArea*) | Accumulated area of all connected grid cells of the LDD from springs (start; lowest values) to mouth (end; highest values) | *LDD*; *pixarea* | PCRaster Accuflux function (Karssenberg et al., 2010) |
| Standard deviation of elevation (*elvstd*) | Amount of elevation variation within a grid cell | MERIT DEM | Upscaling (spatial) with standard deviation |
| Gradient (*gradient*) | Elevation gradient between two connected grid cells | MERIT DEM; *LDD* | $gradient = \frac{\text{abs}(elv_{uc}-elv_{dc})}{D_{uc,dc}}$, where *elv* – elevation, *uc* and *dc* – upstream and downstream cell, $D_{uc,dc}$ – distance between upstream and downstream cells |
| Channel bottom width (*chanbw*) | Width of the bottom of the channel | CaMa-Flood (width); *upArea* | Recomputing zero and negative values based on equation* $chanbw = upArea \cdot 0.0032$ |
| Channel length (*chanlength*) | Length of river channel in each grid cell (can exceed grid-size to account for meandering river) | CaMa-Flood (rivlen) | No transformation was carried out |
| Channel gradient (*changrad*) | Gradient (slope) of river channel inside a grid cell | MERIT DEM; *LDD*; *chanlength* | $changrad = \frac{\text{abs}(elv_{uc}-elv_{dc})}{chanlength_{uc}}$, where *elv* – elevation, *uc* and *dc* – upstream and downstream cell; Note: LDD is used to define *uc* and *dc* |
| Manning's roughness coefficient for channels (*chanman*) | Manning's roughness coefficient of river channel for each grid cell | MERIT DEM; *upArea* | Transformation based on equation* $chanman = 0.25 + 0.015 \cdot \min\left(\frac{50}{upArea_{km^2}}, 1\right) + 0.030 \cdot \min(\frac{elv_m}{2000}, 1)$, where *elv* – elevation, $km^2$ and *m* – values in $km^2$ and m |
| Channel mask (*chan*) | Channel presence in the grid cell indicator. Note LISFLOOD specific requirement to have channels in every 'mask' grid cell | 'mask' (main model's field) | Channel mask is equal to 1 everywhere |
| Side slope (*chans*) | Slope of river banks (i.e. horizontal distance divided by vertical distance) | | Side slope of all channels is 45°, hence side slope is equal to 1 everywhere |

| Bankfull channel depth (*chanbnkf*) | Channel depth (i.e. river bed depth) | *upArea* | Transformation based on equation* $chanbnkf = 0.27 \cdot upArea_{km^2}^{0.33}$, where $km^2$ – values in km$^2$ |
|---|---|---|---|

## 3.3 Regional examples

Most fields in catchment morphology and river network category are quite technical and hard to interpret. The ones that can be easy digested are upstream area and standard deviation of elevation which are presented in Figure 3 for Po River area in 1 arc min and 3 arc min resolution, and in Figure 4 for Amazon River and Brahmaputra River areas at 3 arc min resolution. The field of standard deviation of elevation shows high level of detail over the Brahmaputra River and the benefit of high resolution dataset is clearly seen over the Po River.

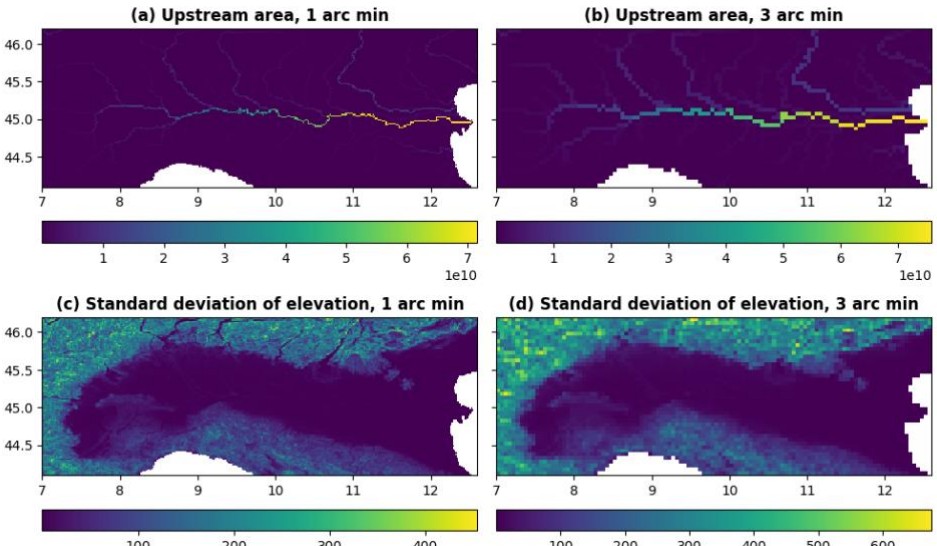

**Figure 3. Upstream drainage area in square meters (upper row, plots a and b) and standard deviation of elevation in meters (lower row, plots c and d) at 1 arc min (~1.9 km at the equator, left column, plots a and c) and 3 arc min (~5.6 km at the equator, right column, plots b and d) resolution for Po River area in Italy.**

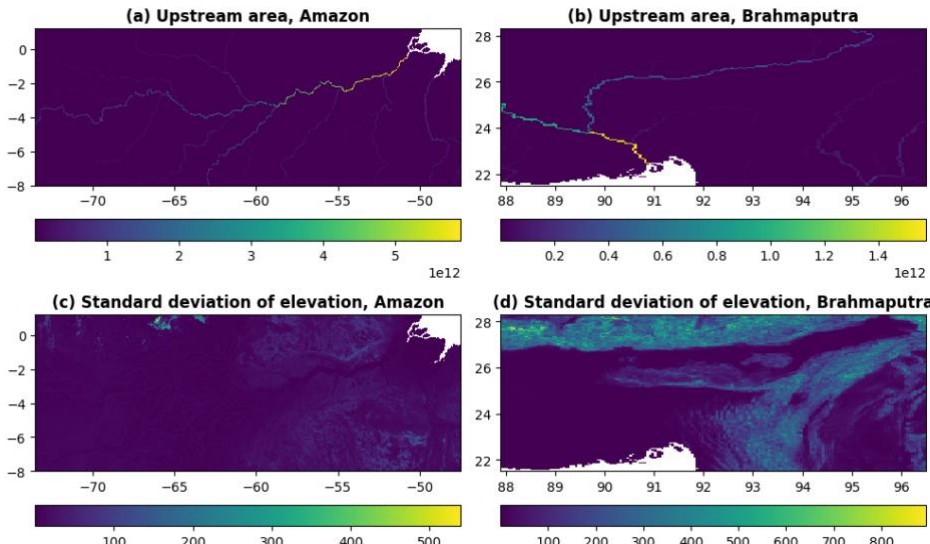

**Figure 4. Upstream drainage area in square meters (upper row, plots a and b) and standard deviation of elevation in meters (lower row, plots c and d) at 3 arc min (~5.6 km at the equator) resolution for Amazon River area (left column, plots a and c) and Brahmaputra River area (right column, plots b and d).**

 **4 Land use fields**

 **4.1 General information**

Land use is an essential component of environmental models. Many models use a sub-grid cell approach where a
single grid cell can include several different land uses with each land use being subject to different prominent
physical processes. This approach allows to keep a high level of accuracy when representing how different types
of land cover affect e.g. the hydrological cycle (e.g. evaporation is different in urban areas compared to forests)
while limiting the increase in computational time. Application of land surface fractions include grid cell weighted
average skin temperature calculations, biogenic flux calculations, urban planning, and climate mitigation plan
preparation. For example, sealed surface fraction is necessary for carbon budget calculations and trace gas
emissions in general, more explicitly for anthropogenic and residential emission calculations, and irrigated crop
and irrigated rice fractions (combined with rice planting and harvesting days) useful for crop yield and methane
emissions modelling.
The dataset differentiates between six different land uses (name in brackets in italics correspond to the field's
name in the data repository):
• Forest: areas where the main hydrological processes are canopy interception, evapotranspiration from
canopies, canopies drainage and evapotranspiration, root uptake and evaporation from the soil (fraction
of forest; *fracforest*, dimensionless fraction);
• Sealed surface: impervious areas where there is no water infiltration into the soil, i.e. water is
accumulated in the surface depression, yet evaporates, but once the depression is full, water is transported
by a surface runoff (fraction of sealed surface; *fracsealed*, dimensionless fraction);
• Inland water: open water bodies where the most prominent hydrological process is evaporation (fraction
of inland water; *fracwater*, dimensionless fraction);
• Irrigated crops: areas used by agriculture – water is abstracted from ground water and surface water
bodies to irrigate the fields. The main hydrological processes connected with the irrigated crops are
canopy interception, evapotranspiration from canopies, canopies drainage and evapotranspiration, root
uptake and evaporation from the soil (fraction of all irrigated crops, excluding rice; *fracirrigated*,
dimensionless fraction);
• Irrigated rice: areas used to grow rice with flooded irrigation agricultural technique, when water is
abstracted from the inland water bodies and delivered to the rice fields. The main hydrological processes
connected with rice fields are soil saturation, flooding, rice growing phase, soil drainage phase (fraction
of irrigated rice; *fracrice*, dimensionless fraction);
• Other land cover: used in canopy interception, evaporation from the canopies, canopy drainage, plant
evapotranspiration, evaporation from the soil hydrological processes. The relative importance of these
processes depends on the LAI (fraction of other cover types; *fracother*, dimensionless fraction).

**4.2 Reference data and methodology**
In models explicitly accounting for sub-grid variability, the fraction of each land use in every cell must be provided
so that process representation for each land use can be weighted accordingly. Here the majority of land use fields
are derived from **The Copernicus Global Land Service (CGLS) Land Cover (LC) 100m map** (further referred
as CGLS-LC100) – a set of global land cover maps at 100 m resolution covering land and ocean area from 90 N
to 60 S, representative of the year 2015; rest of the land use fields (i.e. irrigated crops and irrigated rice fractions)
are derived from (i) **The Spatial Production Allocation Model (SPAM) – Global Spatially-Disaggregated**
**Crop Production Statistics Data for 2010 v2.0** (further referred as SPAM2010) – a global dataset with crop
distribution and production information at 10 km (5 arc min) resolution covering land area from 90 N to 60 S,
representative of the year 2010, and (ii) **The Coordination of Information on the Environment (CORINE)**
**Land Cover (CLC) inventory for 2018** (further referred as CLC2018) – a set of maps describing the land cover/
land use status at 100 m resolution covering land area over Europe (i.e. 39 countries), representative of the time
period 2017-2018 (for reference data details see Appendix 1). The derivation of fractions of the five land use
classes used in LISFLOOD (and additional ocean fraction for consistency check) each follow specific steps (see
Figure 5) summarised in Table 2. Note that LISFLOOD requires all 'mask' (main model's field) grid cells to have
at least one non-zero fraction type, hence the extra step in the generation of the inland water fraction field was to
set empty grid cells (i.e. grid cells that based on the data source are fully covered with ocean) as fully covered
with inland water.

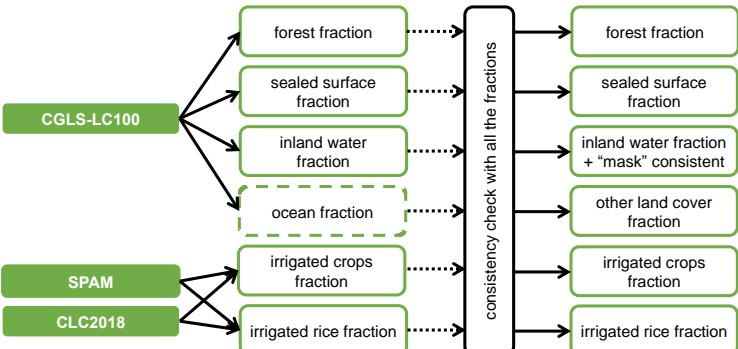

**Figure 5. Workflow of complex manipulations to create land use fields; solid arrows indicate a function transformation, dotted – upscaling; dashed boxes indicate the intermediate fields used for other field generation.**

**Table 2. Fraction of land use fields, their description, data source and applied transformations; 'sum' refers to the sum of all fractions except 'other land cover fraction'; cells with bold italics show required intermediate fields; name in brackets in italics next to each field correspond to the name in the data repository.**

| Field type | Description | Data source (variable) | Transformation (in order) |
|---|---|---|---|
| Forest fraction (*fracforest*) | Evergreen and deciduous needle leaf and broad leaf tree areas | CGLS-LC100 (tree-coverfraction) | Unit conversion % to fraction; Reprojecting and upscaling to final grid and resolution with mean; Consistency check with other fractions |
| Sealed surface fraction (*fracsealed*) | Urban areas, characterizing the human impact on the environment | CGLS-LC100 (urban-coverfraction) | Unit conversion % to fraction, scaled by 0.75[1]; Reprojecting and upscaling to final grid and resolution with mean; Consistency check with other fractions |
| Inland water fraction (*fracwater*) | Rivers, freshwater and saline lakes, ponds and other permanent water bodies over the continents | CGLS-LC100 (water-permanent-coverfraction) | Force Fox Basin and Caspian Sea to be fully covered with water; Unit conversion % to fraction; Reprojecting and upscaling to final grid and resolution with mean; Consistency check with other fractions; Cross-checking with 'mask' and forcing empty grid cells as inland water |
| Irrigated crops fraction (*fracirrigated*) | Irrigated areas of all possible crops excluding rice | SPAM (spam2010v1r0_global_physical-area_CROP_i, 41 crops rice excluding) | Shapefile gridding to its native resolution (~10 km); Unit conversion ha to fractions; Reprojecting and downscaling to CLC2018 grid and resolution (~100 m) with nearest neighbour |
|  |  | CLC2018 (landcover = '212') | Unit conversion class to fraction |
|  |  |  | Merging SPAM- and CLC2018-derived fractions, priority to CLC2018; Reprojecting and upscaling to final grid and resolution with mean; Consistency check with other fractions |
| Irrigated rice fraction (*fracrice*) | Irrigated areas of rice | SPAM (spam2010v1r0_global_physical-area_RICE_i) | Shapefile gridding to its native resolution (~10 km); Unit conversion ha to fractions; Reprojecting and downscaling to CLC2018 grid and resolution (~100 m) with nearest neighbour |
|  |  | CLC2018 (landcover = '213') | Unit conversion class to fraction |
|  |  |  | Merging SPAM- and CLC2018-derived fractions, priority to CLC2018; Reprojecting and upscaling to final grid and resolution with mean; Consistency check with other fractions |

---

[1] For the sealed surface fraction, it is assumed that water can infiltrate in roughly 25 % of urban areas at kilometre scale through e.g. trees along the road, bushes along the fence, grass or moss between concrete tiles or cobble stones.

| | | | |
|---|---|---|---|
| Other land cover fraction (*fracother*) | Agricultural areas, non-forested natural area, pervious surface of urban areas | Non-negative residual from 1 subtracting 'sum' of all other fractions | $fracother = \max((1-sum),0)$ |
| ***Ocean fraction*** (*fracocean*) | Oceans | CGLS-LC100 (discrete_classification = '200') | Unit conversion class to fraction; Forcing NoData to zero over 'mask' grid cells, otherwise – fully covered; Reprojecting and upscaling to final grid and resolution with mean; Consistency check with other fractions |


To ensure consistency between fractions, the sum of all fraction fields must be 1 at any resolution. When sum is
greater than 1, the inland water fraction value is assumed correct (input data corrected prior computation over Fox
Basin and Caspian Sea) and all other fractions are corrected ($fracXX$) following Eq. (1):
$$fracXX = fracXX_{raw}\left(1 - \frac{fracwater_{raw}+fracocean_{raw}+fracforest_{raw}+fracsealed_{raw}+fracirrigated_{raw}+fracrice_{raw}-1}{fracforest_{raw}+fracsealed_{raw}+fracirrigated_{raw}+fracrice_{raw}}\right), \qquad (1)$$
where *raw* refers to the original (i.e. before consistency check) fraction of *XX* which can be the forest, irrigated
crops, rice and sealed surfaces.
The generated fraction fields, e.g. forest (see Figure 6a) and other land cover (see Figure 6b), have generally good
consistency with other up-to-date products like ESA CCI Land Cover time-series v2.0.7 (ESA CCI map viewer
https://maps.elie.ucl.ac.be/CCI/viewer/; Defourny et al., 2017).

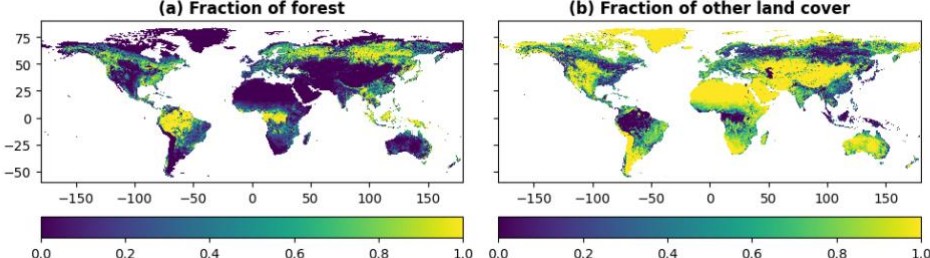

**Figure 6. Fraction of forest (left column, plot a) and fraction of other land cover (right column, plot b) at 3 arc min**
**(~5.6 km at the equator) resolution for global region.**
**4.3 Regional examples**
All fields in land use category are easy to interpret as they represent the fraction of grid cell covered by one or
another surface cover type. The most interesting ones are fraction of forest, fraction of inland water, fraction of
irrigated crops, and fraction of rice which are presented in Figure 7 for Po River area in 1 arc min and 3 arc min
resolution, and in Figure 8 for Amazon River and Brahmaputra River areas at 3 arc min resolution. With high
level of detail visible for the fields of fraction of forest and fraction of inland water (e.g. Amazon River) especially
at the highest spatial resolution (Po River).

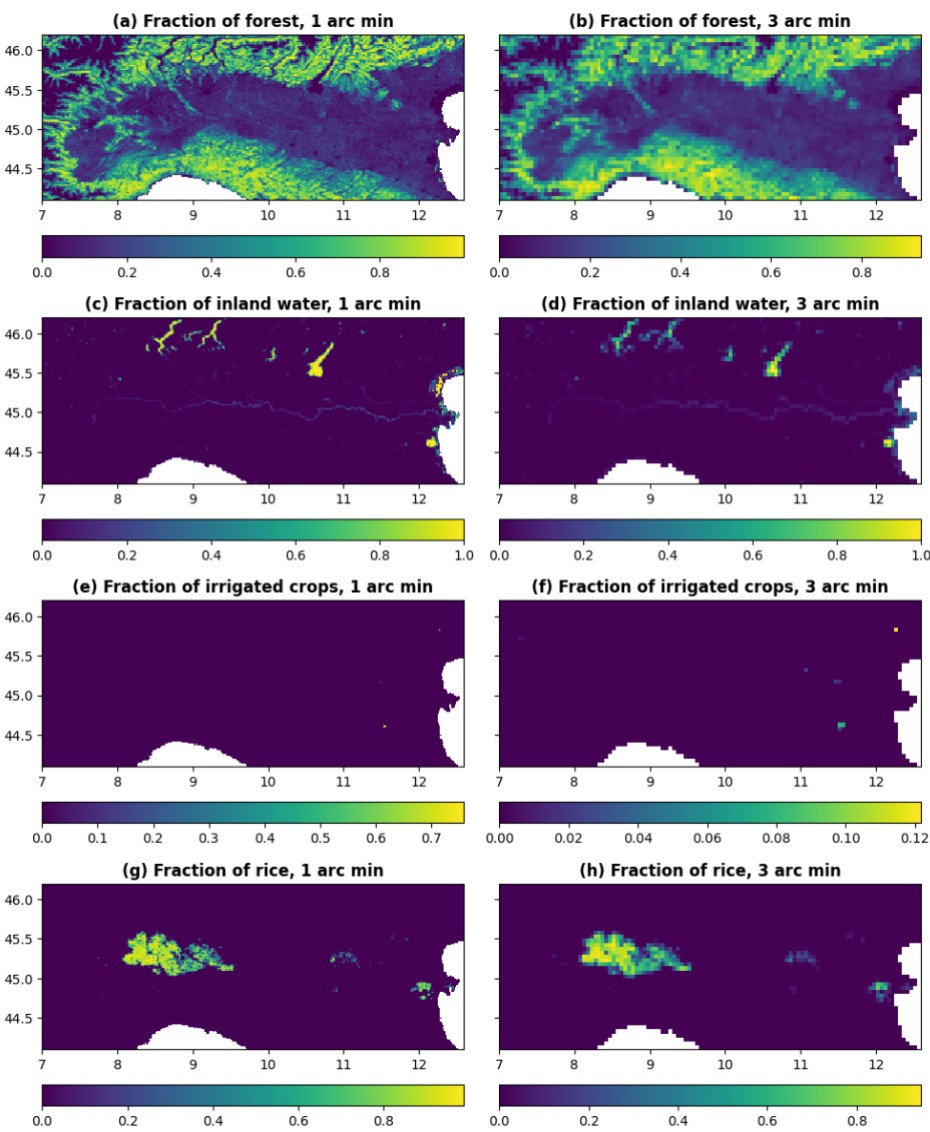

Figure 7. Fraction of forest (upper row, plots a and b), fraction of inland water (second row, plots c and d), fraction of irrigated crops (third row, plots e and f), and fraction of rice (lower row, plots g and h) at 1 arc min (~1.9 km at the equator, left column, plots a, c, e and g) and 3 arc min (~5.6 km at the equator, right column, plots b, d, f and h) resolution for Po River area in Italy.

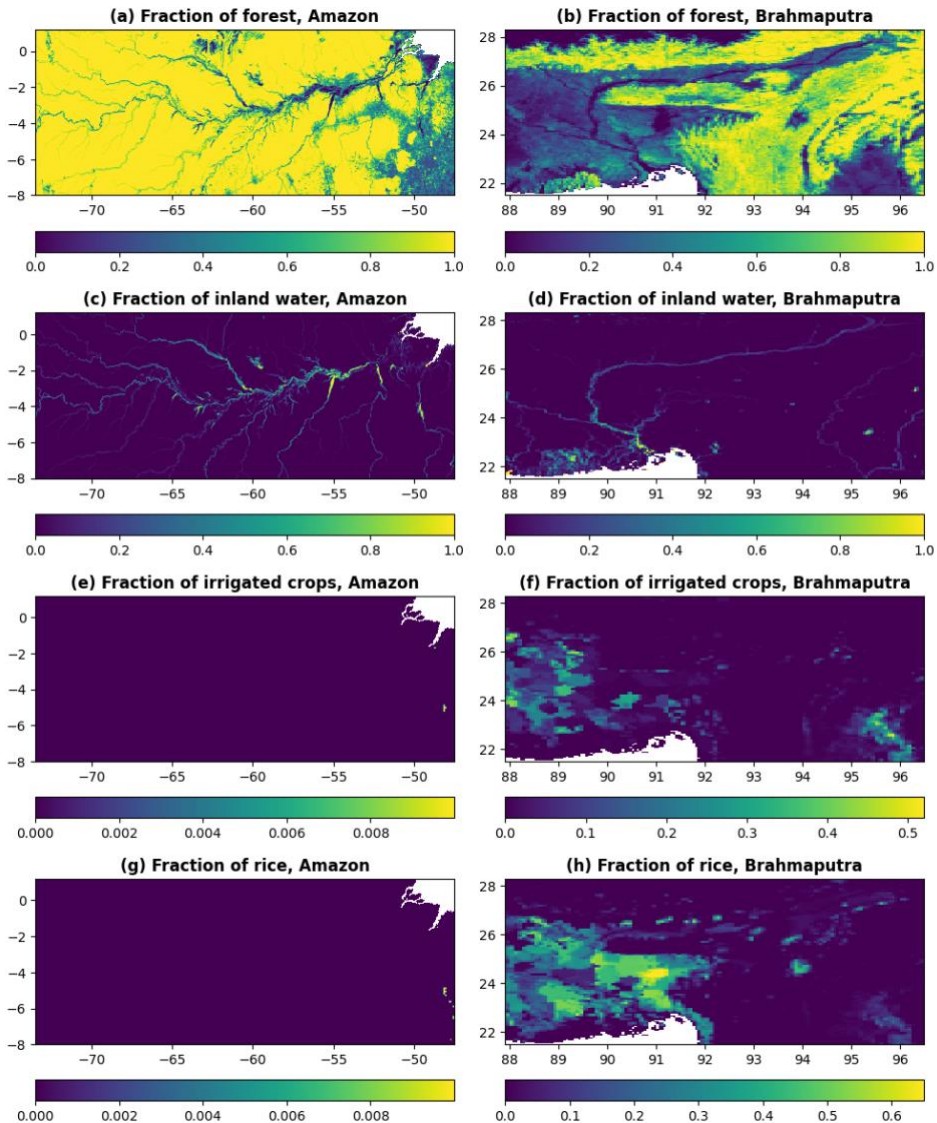

**Figure 8. Fraction of forest (upper row, plots a and b), fraction of inland water (second row, plots c and d), fraction of**
**irrigated crops (third row, plots e and f), and fraction of rice (lower row, plots g and f) at 3 arc min (~5.6 km at the**
**equator) resolution for Amazon River area (left column, plots a, c, e and g) and Brahmaputra River area (right column,**
**plots b, d, f and h).**
**5 Vegetation properties**
**5.1 General information**
Vegetation-related information contributes to the computation of precipitation interception, evaporation,
transpiration, and root water uptake. Depending on the model, vegetation dynamics can be represented with
different degrees of complexity including in hydrology processes, vegetation growth and feedback on climate
(Bonan et al., 2002). Rice being the world's most important food crop and having specific water demands, its
water cycle is often considered explicitly, with planting and harvesting dates being critical information to represent
the inter-annual variability in its water demand, provided the maximum three growing seasons. The variables
allow to model how vegetation affects the hydrology, with a particular focus on root water uptake and transpiration
depending on vegetation type and vegetation state (e.g. water stress conditions). For example, the crop group
number depends on the critical amount of soil moisture below which water uptake from plants is reduced as they
start closing their stomata. Alternative use of fields such as the Leaf Area Index LAI include biomass allocation,
which can be used for fire danger forecasting, and carbon stock monitoring, whilst rice planting/ harvesting days
are important for yearly cycle of methane modelling.

The dataset describes vegetation properties through four variables (note that LAI consists in total of 36 10-day average fields) for each of forest (_f), irrigated crops (_i) and other land cover types (_o), and another six (two types times three seasons) variables for rice (name in brackets in italics correspond to the field's name in the data repository):

- Transpiration rate: crop coefficient (*cropcoef_f*, *cropcoef_i*, *cropcoef_o*, dimensionless);
- Water uptake: crop group number (*cropgrpn_f*, *cropgrpn_i*, *cropgrpn_o*, dimensionless);
- Surface runoff generation and water routing: Manning's surface roughness coefficient (*mannings_f*, *mannings_o*, $s/m^{1/3}$), rice planting and harvesting days (*riceplantingday1*, *riceplantingday2*, *riceplantingday3*, calendar day number; *riceharvestday1*, *riceharvestday2*, *riceharvestday3*, calendar day number);
- Water interception and evaporation: leaf area index (*laif*, *laii*, *laio*, $m^2/m^2$).

## 5.2 Reference data and methodology

In complement to the land use fraction, the distribution of vegetation type and characteristics is required to capture the difference in environmental processes such as water intake of evaporation to be represented accurately. Here the vegetation properties are derived from many data sources using maps to account for the species spatial distribution (i.e. CGLS-LC100 and SPAM2010) and tables to obtain associated hydro-dynamics properties for crops (i) **The Food and Agriculture Organisation (FAO) of the United Nations Irrigation and Drainage Paper No. 56** (further referred as FAO56) – a publication covering geographically referenced statistics for crop development stages, crop coefficients, crop height, rooting depth, and soil water depletion fraction for common crops found across the world, (ii) **Burek** et al. (2014) – a publication covering summarised information for crop coefficients, rooting depth, crop group number and Manning's surface roughness coefficient for different surface types, (iii) **Intara** et al. (2018) – a publication covering oil palm roots architecture, and (iv) **The Wofost 6.0 crop simulation model description** (further referred as SUPIT) – a publication covering crop group information for several crops as examples, and relation of a crop group from water depletion fraction; for river hydraulics **The Open-Channel Hydraulics manual** (further referred as CHOW) – a publication containing information on roughness coefficient over different surfaces. Time evolution of vegetation is based on **The Copernicus Global Land Service (CGLS) Leaf Area Index (LAI) 1km Version 2 collection** (further referred as CGLS-LAI) – a set of global maps without missing data describing vegetation dynamics at 10-day intervals at 1 km resolution covering land area from 90 N to 60 S and representative of the 10-year period of 2010-2019; time evolution of crops is based on **The RiceAtlas v3** (further referred as RiceAtlas) – a spatial database of global rice calendars and production at 1 km resolution for the national production totals to match the years 2010-2012 (for reference data details see Appendix 1). This requires assumptions to be made in case different sources did not contain the same information, and transformations to be applied depending on the vegetation type. The main data sources and general transformation steps (see Figure 9) to derive the 18 vegetation properties fields are summarised in Table 3 and following text. Note that 'crop group number' variable corresponds to a water depletion value and can be averaged across different crop types.

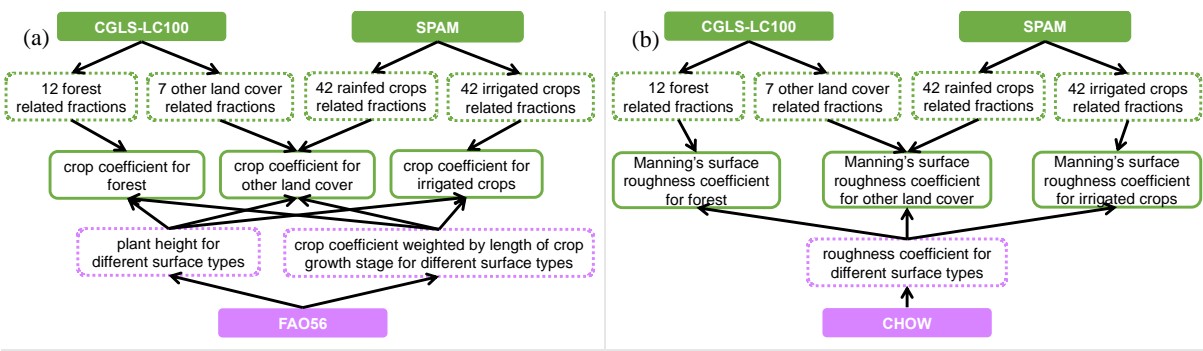

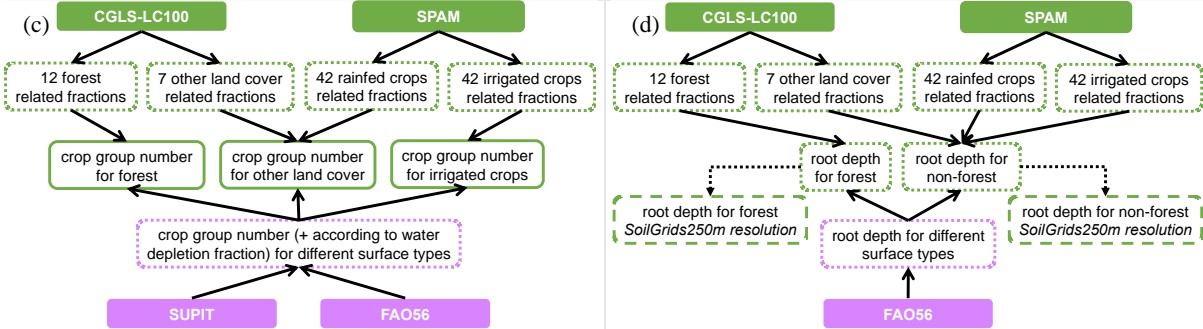

**Figure 9. Workflow of complex manipulations to create some of the vegetation property fields, e.g. crop coefficient (left**
**column, upper row, plot a), Manning's surface roughness coefficient (right column, upper row, plot b), crop group**
**number (left column, lower row, plot c), root depth (right column, lower row, plot d); solid arrows indicate a function**
**transformation, dotted – upscaling; dashed boxes indicate the intermediate fields used for other field generation, dotted**
**– the fields only used for the vegetation-related fields.**
**Table 3. Vegetation property fields, their description, data source and applied transformations; cells with bold italics**
**show required intermediate fields; name in brackets in italics next to each field correspond to the name in the data**
**repository.**

| *Field type* | *Description* | *Data source* | *Transformation (in order)* |
|---|---|---|---|
| Crop coefficient for forest, irrigated crops and other land cover type (*cropcoef_f*, *cropcoef_i*, *cropcoef_o*) | Ratio between the potential (reference) evapotranspiration rate, in mm/day, and the potential evaporation rate of a specific crop (averaged by time and ecosystem type) | CGLS-LC100 (discrete_classification = '111', '112', '113', '114', '115', '116', '121', '122', '123', '124', '125', '126' [forest types], '20', '30', '40', '60', '70', '90', '100' [other land cover types]) | Force Fox Basin and Caspian Sea to be fully covered with water; Unit conversion class to fraction (in total 12 forest related and 7 other land cover related fraction fields); Reprojecting and upscaling to final grid and resolution with mean |
| | | SPAM (spam2010v1r0_global_physical-area_CROP_i/r, 42 crops, 'i' – irrigated, 'r' – rainfed) | Shapefile gridding to its native resolution (~10 km); Unit conversion ha to fractions (in total 42 irrigated crop related and 42 rainfed crop related fraction fields); Reprojecting and downscaling to final grid and resolution with nearest neighbour; Limiting values to 0.0-1.0 interval |
| | | FAO56 (Table 11, 12 – information on crop coefficient and crop height); Intara et al. (2018); Burek et al. (2014) | Average crop coefficient value across climate zones for each crop growing stage and crop/ land cover type; Weighted average of crop coefficient per different crop growth stages (weighted by stage duration in days if available, otherwise mean); Average crop height value across climate zones for each crop/ land cover type |
| | | | Weighted average of relevant crop coefficient for forest, irrigated crops and other land cover type (weighted by crop height and fraction) following Eq. (2); Note: for other land cover type computation of crop coefficient of all rainfed crops is used for CGLS-LC100 (discrete_classification = '40'); Zero/ NoData filling with global mean |
| Crop group number for forest, irrigated crops and other land cover type (*cropgrpn_f*, *cropgrpn_i*, *cropgrpn_o*) | Represents a vegetation type and is an indicator of its adaptation to dry climate (averaged by ecosystem type) | CGLS-LC100 (discrete_classification = '111', '112', '113', '114', '115', '116', '121', '122', '123', '124', '125', '126' [forest types], '20', '30', '40', '60', '70', '90', '100' [other land cover types]) | Same steps as for crop coefficient |
| | | SPAM (spam2010v1r0_global_physical- | Same steps as for crop coefficient |

| | | | |
|---|---|---|---|
| | | area_CROP_i/r, 42 crops, 'i' – irrigated, 'r' – rainfed) | |
| | | FAO56 (Table 22 – information on crop depletion fraction); SUPIT (Table 6.1, 6.2 – information on crop groups); Burek et al. (2014) | Applying function (SUPIT) to water depletion fraction (FAO56) for each crop/ land cover type $cropgrpn = 10 \cdot fr_{dep} - 1.5$, where $fr_{dep}$ – water depletion fraction; Limiting values to 1.0-5.0 interval; Note: if $fr_{dep}$ missing – using precomputed crop group number (Burek et al., 2014) |
| | | | Same steps as for crop coefficient, but in Eq. (2) weighted by fraction only |
| Manning's surface roughness coefficient for forest and other land cover type (*mannings_f*, *mannings_o*) | Roughness or friction applied to the flow by the surface on which water is flowing (averaged by ecosystem type) | CGLS-LC100 (discrete_classification = '111', '112', '113', '114', '115', '116', '121', '122', '123', '124', '125', '126' [forest types], '20', '30', '40', '60', '70', '90', '100' [other land cover types]) | Same steps as for crop coefficient |
| | | SPAM (spam2010v1r0_global_physical-area_CROP_i/r, 42 crops, 'i' – irrigated, 'r' – rainfed) | Same steps as for crop coefficient |
| | | CHOW (Table 5, 6 – information on roughness coefficient n); Burek et al. (2014) | Matching roughness coefficient for each crop/ land cover type |
| | | | Same steps as for crop coefficient, but in Eq. (2) weighted by fraction only |
| Leaf area index for forest, irrigated crops and other land cover type (*laif*, *laii*, *laio*) | Defined as half the total area of green elements of the canopy per unit horizontal ground area $m^2/m^2$ (10-day average; 36 fields in total) | CGLS-LAI 10-day average for 2010-2019; *fracforest*; *fracirrigated*; *fracother* | Upscaling to final temporal resolution (in total 36 LAI fields); Reprojecting and upscaling to final grid and spatial resolution with unweighted mean; Filtering sparce areas of relevant fractions $fr < 0.7$, where $fr$ – fraction; NoData filling DEEP (upscaling to 1, 3, 15 arc min, 1, 3, 15, 60 degrees spatial resolution with unweighted mean; replacing NoData at final resolution with first available precomputed less coarser resolution, if not – with zero) |
| Rice planting day (*riceplantingday1*, *riceplantingday2*, *riceplantingday3*) | Most probable day of the year when rice is planted for the first, second and third time | RiceAtlas (PLANT_PKn, 3 seasons) | Ordering planting seasons by increasing Julian day (in total 3 planting dates per spatial unit); Shapefile gridding to final grid and spatial resolution (in total 3 fields); Note: if less than 3 seasons – repeating last available planting/ harvesting seasons date; NoData filling with global unweighted mode date of first planting/ harvesting season (i.e. 105 – 15th April/ 227 – 15th August) |
| Rice harvest day (*riceharvestday1*, *riceharvestday2*, *riceharvestday3*) | Most probable day of the year when rice is harvested after planting for the first, second and third time | RiceAtlas (HARV_PKn, 3 seasons) | |
| ***Root depth for forest and non-forest*** (*root_depth_f*, *root_depth_o*) | Deepest soil depth reached by the crop roots | CGLS-LC100 (discrete_classification = '111', '112', '113', '114', '115', '116', '121', '122', '123', '124', '125', '126' [forest types], '20', '30', '40', '60', '70', '90', '100' [other land cover types]) | Same steps as for crop coefficient |
| | | SPAM (spam2010v1r0_global_physical-area_CROP_i/r, 42 crops, 'i' – irrigated, 'r' – rainfed) | Same steps as for crop coefficient |

| | | FAO56 (Table 22 – information on crop rooting depth); Burek et al. (2014) | Matching rooting depth for each crop/ land cover type |
|---|---|---|---|
| | | | Same steps as for crop coefficient, but in Eq. (2) weighted by fraction only; Downscaling to native SoilGrids250m resolution with nearest neighbour (for soil depth calculations) |


The final step of the crop coefficient, crop group number, Manning's surface roughness coefficient, and additional
crop height (for crop coefficient calculation) and root depth (for soil depth calculation, see Section 6.2) for forest,
irrigated crops and other land cover type is to compute weighted average of their components (e.g. different forest
types) following Eq. (2):
$$K = \frac{A_1 \cdot fr_1 \cdot K_1 + A_2 \cdot fr_2 \cdot K_2 + \cdots + A_N \cdot fr_N \cdot K_N}{A_1 \cdot fr_1 + A_2 \cdot fr_2 + \cdots + A_N \cdot fr_N},$$ (2)
where $A$ is a scaling parameter (equals 1, except for crop coefficient where it equals to crop height), $fr$ refers to
the fraction of crop or land cover type, $K$ – default (i.e. source table based) variable in question values, $1..N$ –
number of crop or land cover types included in the field (i.e. for forest $N=12$, irrigated crops $N=41$, other land
cover type $N=7$ and for CGLS-LC100 type '40' (cropland) default values are based on 42 rainfed crops).
The generated vegetation property fields, e.g. crop coefficient for forest (see Figure 10a) and other land cover (see
Figure 10b), follow main features of e.g. generated forest fraction.

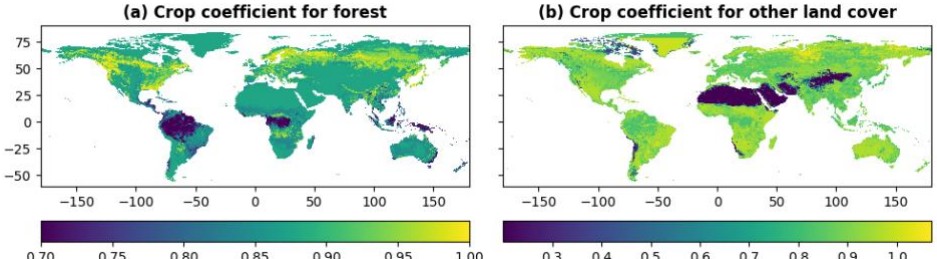

**Figure 10. Crop coefficient for forest (left column, plot a) and crop coefficient for other land cover type (right column,**
**plot b) at 3 arc min (~5.6 km at the equator) resolution for global region.**
**5.3 Regional examples**
All fields in the vegetation properties category are complementary to the land use fractions, and help to understand
for example the difference in evaporation water intake. The fields easiest to interpret are the crop coefficient and
the crop group number which are presented for forest in Figure 11 for Po River area in 1 arc min and 3 arc min
resolution, and in Figure 12 for Amazon River and Brahmaputra River areas at 3 arc min resolution. For example,
fields of crop group number for forest (i.e. different forest type) show transition of vegetation resilience towards
dry conditions in the Brahmaputra River area.

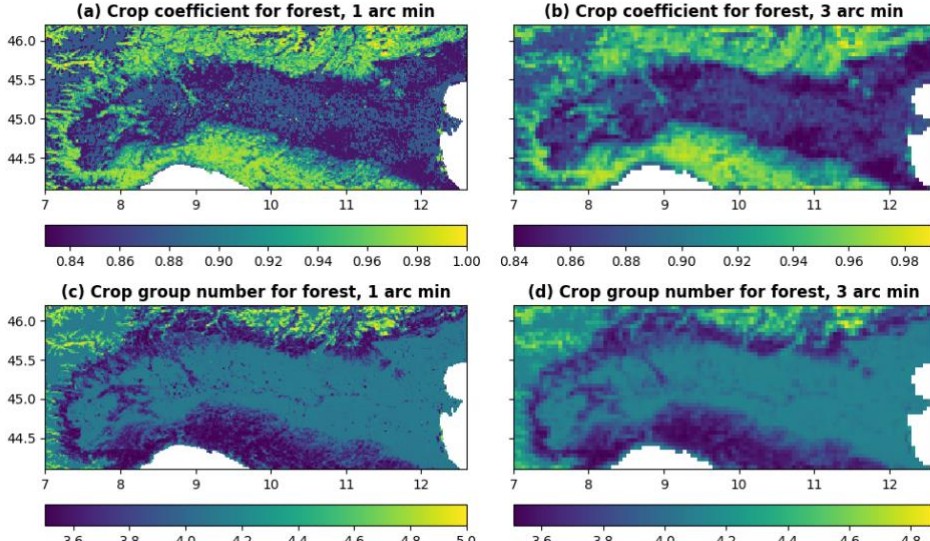


**Figure 11. Crop coefficient for forest (upper row, plots a and b) and crop group number for forest (lower row, plots c and d) at 1 arc min (~1.9 km at the equator, left column, plots a and c) and 3 arc min (~5.6 km at the equator, right column, plots b and d) resolution for Po River area in Italy.**

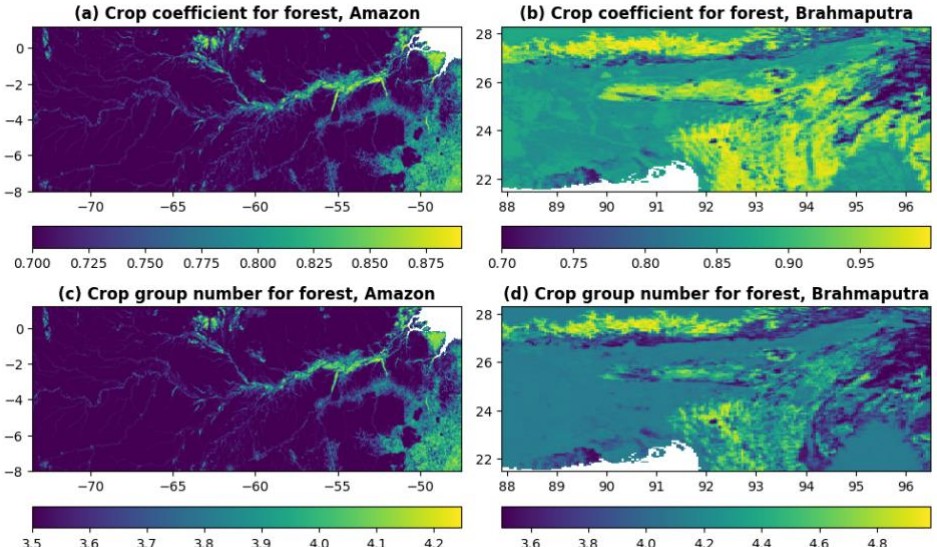

**Figure 12. Crop coefficient for forest (upper row, plots a and b) and crop group number for forest (lower row, plots c and d) at 3 arc min (~5.6 km at the equator) resolution for Amazon River area (left column, plots a and c) and Brahmaputra River area (right column, plots b and d).**

## 6 Soil properties

### 6.1 General information

In land surface and distributed hydrological models, the water movement, storage and plants' water-uptake from the soil are often described by the soil/-water retention curve (SWRC). The SWRC is derived empirically by measuring how water is retained and released by different soil types. Throughout time different SWRC have been developed and integrated into models, the most widely applied are Van Brooks and Corey (Brooks and Corey, 1964), Fredlund and Xing (Fredlund and Xing, 1994), van Genuchten (van Genuchten, 1980), and Gardner (Gardner, 1956) SWRCs. Different SWRC equations require different parameters, some shared between different SWRC concepts, e.g. referring physical soil characteristics such as water saturated and unsaturated content, hydraulic conductivity and pore size, others uniquely describing the SWRC function shape, not directly related to soil properties. Often, for computational reasons, the soil profile from ground level to bedrock depth is sliced into layers, at the modeller's choice, and the SWRC function is applied to each soil layer. Alternative use of soil properties is for soil moisture calculations.

The dataset includes variables required to apply the Van Genuchten SWRC equations (van Genuchten, 1980) to describe the water dynamics through a vertical soil profile composed of three layers (1, 2, 3), each variable is required for each soil layer and for forest (_f) or non-forest (_o) land use, with different soil depth in forest (_f) and non-forest (_o) areas following root depth values from Allen at al. (1998), further referred as FAO56, (total of 29 variables; name in brackets in italics correspond to the field's name in the data repository):

- Soil profile: surface layer depth (*soildepth1_f*, *soildepth1_o*, mm), middle layer depth (*soildepth2_f*, *soildepth2_o*, mm), subsoil depth (*soildepth3_f*, *soildepth3_o*, mm);
- Soil hydraulic properties: saturated (*thetas1_f*, *thetas1_o*, *thetas2_f*, *thetas2_o*, *thetas3*, $m^3/m^3$) and residual (*thetar1*, *thetar2*, *thetar3*, $m^3/m^3$) volumetric soil moisture content, pore size index (*lambda1_f*, *lambda1_o*, *lambda2_f*, *lambda2_o*, *lambda3*, dimensionless), Van Genuchten equation parameter (*genua1_f*, *genua1_o*, *genua2_f*, *genua2_o*, *genua3*, cm$^{-1}$), saturated soil conductivity (*ksat1_f*, *ksat1_o*, *ksat2_f*, *ksat2_o*, *ksat3*, mm/day).

### 6.2 Reference data and methodology

Soil proprieties are derived from **The International Soil Reference and Information Centre (ISRIC) SoilGrids250m global gridded soil information release 2017** (further referred as SoilGrids250m) – an output of special predictions produced by the SoilGrids system, as a set of global soil property and class maps on soil

characteristics at six standard depths, including soil textures (clay, silt, sand), depth to bedrock, bulk density,
organic carbon, pH and cation exchange capacity at 250 meters resolution covering land area with no permanent
ice and representative for the year 2010 (for reference data details see Appendix 1); and are computed for both
forested and non-forested (also known in literature as 'others') areas, expressed as fractions (main source is forest
fraction based on CGLS-LC100, see Section **Error! Reference source not found.**), where non-forested area is
the complementary fraction of forest. Soil depth layers are derived first and used as input to the soil hydraulic
equations used to derive the properties, following a sequential workflow (see Figure 13 and Table 4). Equations
used are from Toth et al. (2015).

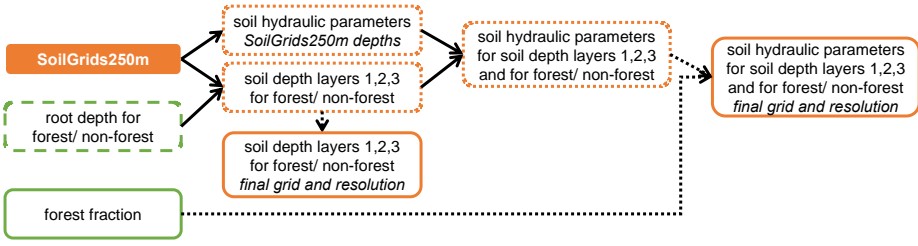

**Figure 13. Workflow to generate the soil related fields; solid arrows indicate a function transformation, dotted –**
**upscaling; dashed boxes indicate the intermediate fields used for other field generation, dotted – the fields only used**
**for the soil-related fields; 'SoilGrids250m depths' – fields at the SoilGrids250m native grid and resolution with six**
**default depths, 'final grid and resolution' – fields at the dataset's final grid and resolution, boxes with no explicit**
**indication – fields at SoilGrids250m native grid and resolution only.**
**Table 4. Soil property fields, their description, and applied transformations; name in brackets in italics next to each**
**field correspond to the name in the data repository.**

| Field type | Description | Data Source | Transformation (in order) |
|---|---|---|---|
| Soil depth layers 1, 2, 3 for forest and non-forest (*soildepth1_f*, *soildepth1_o*, *soildepth2_f*, *soildepth2_o*, *soildepth3_f*, *soildepth3_o*) | Root depths assumed to divide the total soil depth between topsoil (surface [layer 1] and middle [layer 2]) and subsoil (bottom [layer 3]) | SoilGrids250m (absolute_depth_to_bedrock); *root_depth_f*; *root_depth_o* | Transforming at SoilGrids250m native grid and resolution as described in Appendix 3 'Soil Depth' (in total 3 forest and 3 non-forest soil depth layer fields); Reprojecting and upscaling to final grid and resolution with unweighted mean; NoData filling DEEP (upscaling to 1, 3, 15 arc min, 1, 3, 15, 60 degrees spatial resolution with unweighted mean; replacing NoData at final resolution with first available precomputed less coarser resolution, if not – with zero) |
| Saturated volumetric soil moisture content for soil depth layers 1, 2, 3, and for forest and non-forest (*thetas1_f*, *thetas1_o*, *thetas2_f*, *thetas2_o*, *thetas3*) | Saturated water content soil hydraulic property representing the maximum water content in the soil | SoilGrids250m (clay_content, silt_content, bulk_density); *soildepth1_f*; *soildepth1_o*; *soildepth2_f*; *soildepth2_o*; *soildepth3_f*; *soildepth3_o*; *fracforest* | Transforming at SoilGrids250m native grid and resolution as described in Appendix 3 'Soil hydraulic parameters' (in total 5 fields per soil hydraulic parameter, except *thetar* – only 3 as no forest/ non-forest separation); Limiting values and weighting by forest/ non-forest fraction (limits $thetas < 1.0$, $thetar < thetas$, $lambda \leq 0.42$, $genua \leq 0.055$, $ksat > 0.0$); Upscaling to final grid and resolution with unweighted mean; NoData filling DEEP (upscaling to 1, 3, 15 arc min spatial resolution with unweighted mean; replacing NoData at final resolution with first available |
| Residual volumetric soil moisture content for soil depth layers 1, 2, 3 (*thetar1*, *thetar2*, *thetar3*) | Residual water content soil hydraulic property representing the minimum water content in the soil | SoilGrids250m (clay_content, silt_content); *soildepth1_f*; *soildepth1_o*; *soildepth2_f*; *soildepth2_o*; *soildepth3_f*; *soildepth3_o*; *fracforest* | |
| Pore size index for soil depth layers 1, 2, 3, and for forest and non-forest (*lambda1_f*, *lambda1_o*, *lambda2_f*, *lambda2_o*, *lambda3*) | Van Genuchten parameter λ (also referred as 'n-1' in literature) soil hydraulic property representing the pore size index of the soil | SoilGrids250m (clay_content, silt_content, bulk_density, organic_carbon_content); *soildepth1_f*; *soildepth1_o*; *soildepth2_f*; *soildepth2_o*; *soildepth3_f*; *soildepth3_o*; *fracforest* | |

| Van Genuchten equation parameter for soil depth layers 1, 2, 3, and for forest and non-forest (*genua1_f*, *genua1_o*, *genua2_f*, *genua2_o*, *genua3*) | Van Genuchten parameter α soil hydraulic property | SoilGrids250m (clay_content, silt_content, bulk_density, organic_carbon_content); *soildepth1_f*; *soildepth1_o*; *soildepth2_f*; *soildepth2_o*; *soildepth3_f*; *soildepth3_o*; *fracforest* | precomputed less coarser resolution, if not – with global unweighted mean) |
|---|---|---|---|
| Saturated soil conductivity for soil depth layers 1, 2, 3, and for forest and non-forest (*ksat1_f*, *ksat1_o*, *ksat2_f*, *ksat2_o*, *ksat3*) | Saturated hydraulic conductivity soil hydraulic property representing the ease with which water moves through pore spaces of the soil | SoilGrids250m (clay_content, silt_content, soil_pH, cation_exchange_capacity); *soildepth1_f*; *soildepth1_o*; *soildepth2_f*; *soildepth2_o*; *soildepth3_f*; *soildepth3_o*; *fracforest* | |

Two of the most common soil parameters of land surface and hydrological models, saturated hydraulic
conductivity *ksat* and saturated water content, are shown in Figure 14.
Saturated hydraulic conductivity *ksat* (see Figure 14a) ranges from 2 to 7445 mm/day. The highest *ksat* values are
concentrated in desertic areas such as the Sahara, Arabian Peninsula, Gobi, Patagonian, Sonoran-Mojave and
Kalahari and Namib deserts. Low *ksat* between, 2 and 18 mm/day, are found in the Amazon river basin, the lower
Mississippi river basin and South East Asia. *ksat* was visually compared against 8 global datasets developed with
different input data and/ or PTFs (Zhang and Schaap, 2019; Gupta et al., 2021); a general agreement is noticeable
in areas that show low variability across all datasets. Northern Russia, Canada, South East Asia and Sonoran-
Mojave Desert are the areas with high variability among datasets, with values ranging from very low to very high
*ksat*. Source of uncertainties in *ksat* values are primarily due to little availability of soil samples and measurements
carried out in those areas. Moreover, the climatic context plays a relevant role in clay mineralogy composition,
organic composition and soil pores structure (Hodnett and Tomasella, 2002), which influence how water flows
through the soil. Therefore, the PTF developed using soil samples collected in temperate areas (such as Europe)
are expected to have a different hydraulic behaviour compared to those collected in tropical climates (Gupta et
al., 2021), as also seen in Figure 14a.
Saturated water content (see Figure 14b) ranges between 0.27 to 0.79, with 80% of values between 0.40 and 0.46.
A comparison with other global datasets was not carried out, however uncertainties are expected to be of the same
order of magnitude than those of *ksat* given the fact the saturated water content is calculated using bulk density
and clay content data.

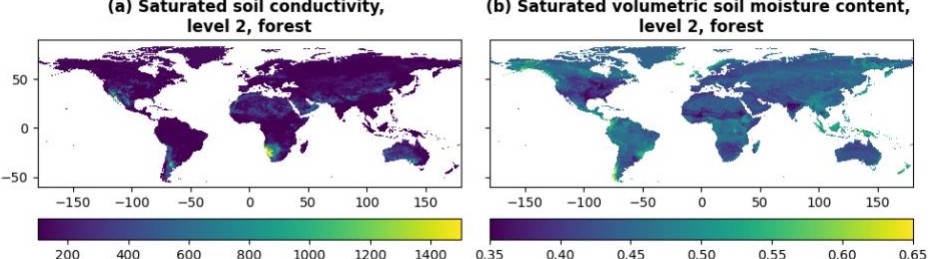

**Figure 14. Saturated soil hydraulic conductivity for forested areas of soil depth layer 2 in mm per day (left column,**
**plot a) and saturated volumetric soil moisture (i.e. water) content for forested areas of soil depth layer 2 (right column,**
**plot b) at 3 arc min (~5.6 km at the equator) resolution for global region.**
**6.3 Regional examples**
The majority of soil properties fields are easy to interpret. Saturated soil conductivity *ksat* and saturated volumetric
soil moisture content are presented for forested areas of soil depth layer 2 in Figure 15 for the Po River area in 1
arc min and 3 arc min resolution, and in Figure 16 for the Amazon River and the Brahmaputra River areas at 3
arc min resolution. The field of saturated soil conductivity for forest shows how easy it is for water to penetrate
soil depending on forest type, and the field of saturated volumetric soil moisture content shows what is the
maximum amount of water that the soil can absorb depending on forest type have interesting features over
Brahmaputra River area.

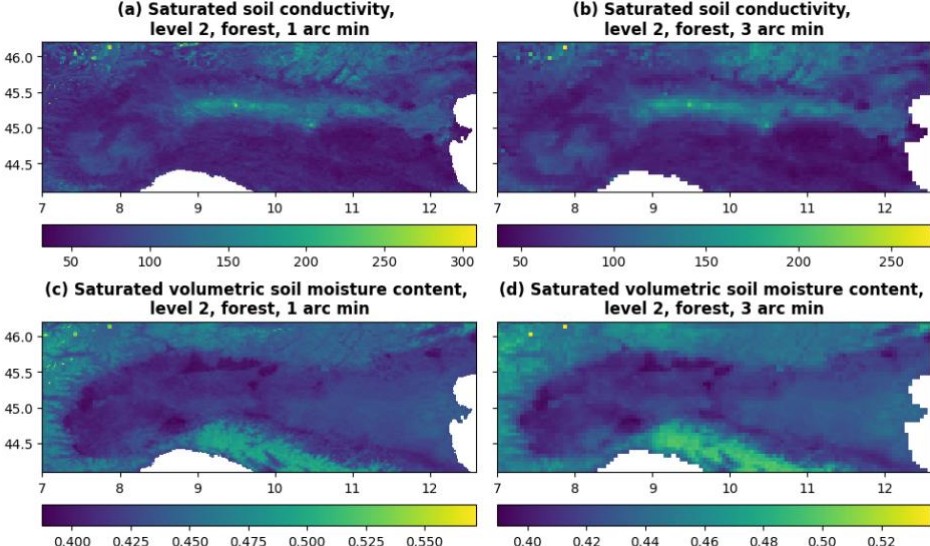

Figure 15. Saturated soil hydraulic conductivity for forested areas of soil depth layer 2 in mm per day (upper row, plots a and b) and saturated volumetric soil moisture (i.e. water) content for forested areas of soil depth layer 2 (lower row, plots c and d) at 1 arc min (~1.9 km at the equator, left column, plots a and c) and 3 arc min (~5.6 km at the equator, right column, plots b and d) resolution for Po River area in Italy.

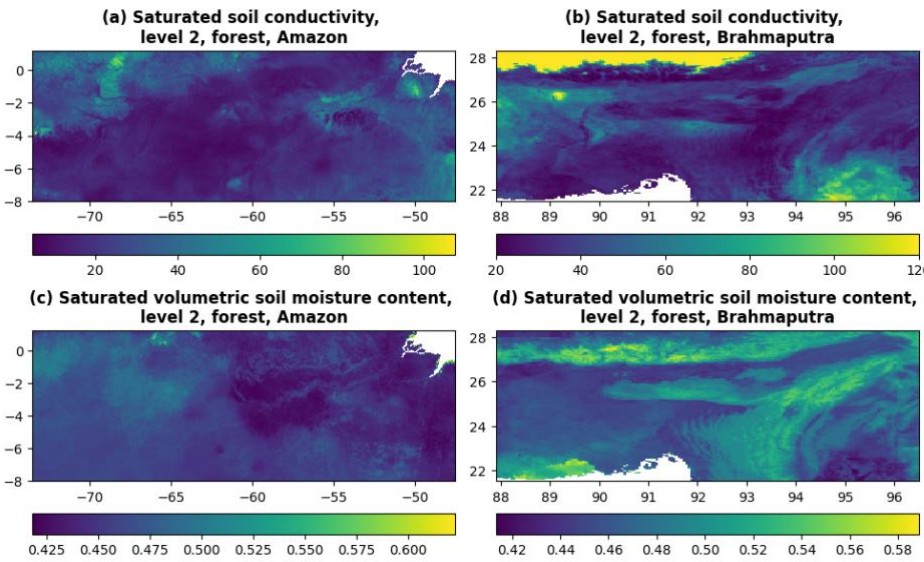

Figure 16. Saturated soil hydraulic conductivity for forested areas of soil depth layer 2 in mm per day (upper row, plots a and b) and saturated volumetric soil moisture (i.e. water) content for forested areas of soil depth layer 2 (lower row, plots c and d) at 3 arc min (~5.6 km at the equator) resolution for Amazon River area (left column, plots a and c) and Brahmaputra River area (right column, plots b and d).

# 7 Lakes

## 7.1 General information

Lakes (and reservoirs) are important as they influence river discharge variability but also the atmosphere regionally and globally. The area covered by lakes can be used for computing evaporation from open water, freshwater storage, unregulated surface water extent, fresh water scarcity indexes, and biogenic green house gas emission, as well as for reproducing different climate mitigation scenarios. The CEMS_SurfaceFields_2022 dataset only includes data on lake extent and not reservoirs (generally smaller), described as a lake mask where the presence of lakes is consistent with fraction of inland water; the field's name in the data repository is *lakemask*, dimensionless).

## 7.2 Reference data and methodology

The lake mask field is derived from **The Global Lakes and Wetlands Database** (further referred as GLWD) – a global database of water bodies at spatial resolutions of up to 1:1 million – GLWD-1 with 3067 largest lake and 654 largest reservoir polygons, and GLWD-2 with ~250000 smaller lake and reservoir polygons (see Table 7).

**Table 5. Lake field, its description, data source and transformation; name in brackets in italics next to the lake field corresponds to the name in the data repository.**

| Field type | Description | Data source | Transformation (in order) |
|---|---|---|---|
| Lake mask (*lakemask*) | Area covered by lakes only (binary representation) | GLWD (GLWD-1, GLWD-2, lake type only); *fracwater* | Filtering non-lake spatial units; Shapefile gridding to final grid and resolution; If *fracwater* > 0 and GLWD is 'lake', then *lakemask* is 1, otherwise 0 |

## 7.3 Regional examples

The lake mask field is easy to interpret as it shows which grid cells from fraction of inland water field have lakes. The lake mask field is presented in Figure 17 for Po River area in 1 arc min and 3 arc min resolution, and in Figure 18 for Amazon River and Brahmaputra River areas at 3 arc min resolution, where it shows the abundance of lakes over Amazon River area and detailed lake shapes over Po River area described by the 1 arc min resolution field.

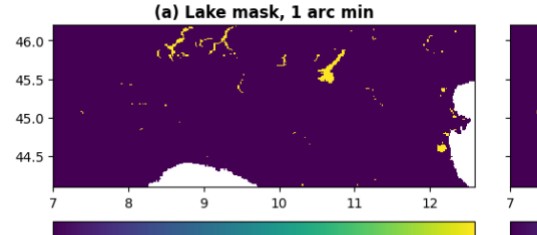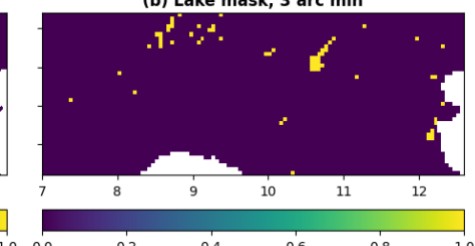

**Figure 17. Lake mask at 1 arc min (~1.9 km at the equator, left column, plot a) and 3 arc min (~5.6 km at the equator, right column, plot b) resolution for Po River area in Italy.**

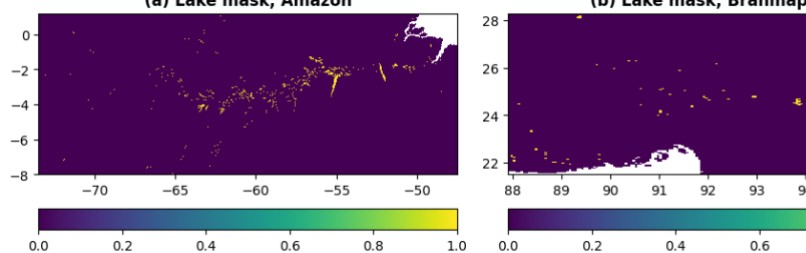

**Figure 18. Lake mask at 3 arc min (~5.6 km at the equator) resolution for Amazon River area (left column, plot a) and Brahmaputra River area (right column, plot b).**

## 8 Water demand

### 8.1 General information

Some environmental models explicitly represent a number of the human interventions impacting on the water cycle. One of the most common is water demand, which represents the withdrawal of water from natural water sources (e.g. rivers, reservoirs, groundwater) to satisfy the water demand for anthropogenic use. The segregation of the total water demand for anthropogenic use into four main sectors, namely domestic, energy, industrial, and livestock water withdrawal, enables a more accurate representation of the processes, and follows the Food and Agriculture Organisation of the United Nations (FAO) terminology (Kohli et al., 2012). Domestic water withdrawal represents indoor and outdoor household water use as well as other uses (e.g. industrial and urban agriculture) connected to the municipal system (e.g., water use by shops, schools, and public buildings). Electricity (energy) water withdrawal is the water use for the cooling of thermoelectric and nuclear power plants. Water withdrawal for industry is the water used for fabricating, processing, washing, cooling or transporting products,

also includes water within the final products and water used for sanitation within the manufacturing facility.
Livestock withdrawal is the demand for drinking and cleaning purposes of livestock.
Higher accuracy in environmental modelling is achieved by differentiating water demand sources and by
allocating different levels of priority to different usages. Within LISFLOOD, for instance, water demand for the
energy sector and flooded irrigation (rice crops) is supplied by surface water bodies only, while non-flooded
irrigation, domestic, industrial, livestock water demand can be supplied by both groundwater and surface water
bodies. Moreover, domestic water demand has the highest priority in case of water scarcity conditions.
It must be noted that the fields of water demand for agriculture are not included in this dataset because LISFLOOD
computes crops water demand internally by accounting for climatic conditions, information on land cover (see
Section 4.2), crops properties (see Section 5.2), and soil properties (see Section 6.2). Conversely, fields
representing the volume of water to satisfy the domestic, energy, industrial, and livestock demand must be
provided as input. Domestic, industrial, energy, and livestock water demand volumes have seasonal (e.g. due to
temperature differences) and inter-annual variations (e.g. due to population changes and different economic
conditions). In order to account for this variability, in LISFLOOD the four sectoral water demand fields provide
daily water demand data with monthly or annual variability from 01.01.1979 to 31.12.2019: the water demand
values are provided in mm/day, one field per month (the first day of each month is used as representative
timestamp for the entire month) for domestic and energy demand, one value per year (the monthly fields are
repeated twelve times per each year) for industrial and livestock demand.
Water availability, ecosystem long term ecological status, and anthropogenic needs must be accounted for to
evaluate the long term sustainability of water withdrawals. However, the spatial scales of water use data and
available water resources data often do not match due to different ways of data surveying and/or modelling
(McManamay et al., 2021; Zhang et al., 2023) and this creates a technical hurdle. Alternative use of the gridded
sectoral water demand information is e.g. for (i) the statistical analysis of long term spatiotemporal patterns and
trends of water demand; (ii) the evaluation of the long term sustainability and impacts of water withdrawals (e.g.
in connection to remote sensing-derived datasets of surface water extent or groundwater total storage); (iii) the
analysis of ecosystem–water–food–energy nexus (Karabulut et al., 2016); (iv) the evaluation of the impacts on
water resources of economical and price policies (Dolan et al., 2021); (v) the analysis of the responses in sectoral
water use during hydro climatic extremes (Belleza et al., 2023).
The CEMS_SurfaceFields_2022 dataset includes water demand for four main sectors (note that each sector
consists in total of 12 daily water demand fields per 41 (1979-2019) years, so 492 fields per sector) for (name in
brackets in italics correspond to the field's name in the data repository): livestock (*liv*, mm/day), industry (*ind*,
mm/day), energy production, (*ene*, mm/day) and domestic use (*dom*, mm/day). The temporal extension of the
water demand fields presented in this manuscript includes the most recent information of water demand at the
time of the dataset's preparation. Readers that are interested in using more recent water demand data are invited
to follow the protocol presented in Section 8.2 to further extend in time the provided fields.
**8.2 Reference data and methodology**
Global gridded water demand fields with monthly variability were generated for the four sectors using the data
sources listed here and following the transformations summarised in Table 8 (for additional information and extra
details see GitHub repository 'lisflood-utilities/src/lisfloodutilities/water-demand-historic at master · ec-
jrc/lisflood-utilities · GitHub', last accessed: 21.01.2024): (i) **AQUASTAT** – the FAO's global information
system with yearly country data on water resources and agricultural water management for "Gross Domestic
Product (GDP)", "Industry, value added to GDP", "Agricultural water withdrawal", "Industrial water withdrawal",
"Municipal water withdrawal", "Total water withdrawal", and "Irrigation water withdrawal"; (ii) **United States**
**Geological Survey National Water Information System** (further referred as USGS NWIS) – a United States
(US) database on water use data for the annual state statistics for "Domestic total self-supplied withdrawals, fresh,
in Mgal/d", "Public Supply total self-supplied withdrawals, fresh, in Mgal/d", "Industrial total self-supplied
withdrawals, fresh, in Mgal/d", "Total Thermoelectric Power total self-supplied withdrawals, fresh, in Mgal/d",
"Total Thermoelectric Power power generated, in gigawatt-hours", and "Livestock total self-supplied
withdrawals, fresh, in Mgal/d"; (iii) **Global Change Analysis Model** (further referred as GCAM) – an integrated,
multi-sector model's output that provides estimates on water withdrawals for energy, agriculture, and municipal
uses as lumped values of 235 hydrologic basins; (iv) **Global-scale gridded estimates of thermoelectric power**
**and manufacturing water use** (further referred as Vassolo and Doll, 2005) – a global-scale gridded on 0.5° by
0.5° grid estimate of water withdrawal for cooling of thermal power stations and for manufacturing, representative
for the year 1995; (v) **The Gridded Livestock of the World (GLW) version3** (further referred as GLW3) – a
global-scale gridded on 0.083333° by 0.083333° (~10 km at the equator) grid of eight livestock species
distribution, representative for the year 2010; (vi) **World Bank manufacturing value added and gross domestic**
**product** (further referred as World Bank) – data provide "Manufacturing, value added (constant 2015 US$)"
values (further referred as MVA) and "Gross Domestic Product GDP (constant 2015 US$)" values; (vii) **The**
**Global Human Settlement Population Grid multitemporal version R2019A** (further referred as GHS-POP) –
a global-scale gridded on 9 arc sec (~300 m at the equator) grid distribution of population, expressed as the number
of people per grid cell, representative for the years 1975, 1990, 2000 and 2015; (viii) **Thematic Mapping**
**Country Borders** shapefile (further referred as TM 'country borders') – world country borders dataset; (ix) **The**
**United States Census Bureau** Cartographic Boundary Files – Shapefile (further referred as US CB) – the State
boundaries for the USA, representative for the year 2018; (x) **Multi-Source Weather** (further referred as MSWX)
– a global-scale gridded high-resolution (3-hourly, 0.1°), bias-corrected meteorological product with 2-meter daily
and monthly maximum and minimum air temperature; (xi) **Huang et al. (2018)** – a publication presenting 0.5°
resolution global monthly gridded sectoral water withdrawal dataset for the period 1971–2010 with calibrated R
coefficient values and technique for temporal downscaling of domestic and energy water demands.
The water demand values are provided in mm/day, one field per month from 01.01.1979 to 31.12.2019 (the first
day of each month is used as the representative timestamp for the entire month). The methodology applied largely
follows Huang et al. (2018), with the key differences being the use of freely available datasets and the higher
resolution of the resulting fields. Spatial downscaling was achieved following the approach of Hejazi et al. (2014);
temporal downscaling was performed following the approaches of Wada et al. (2011), Voisin et al. (2013) and
Huang et al. (2018). It should be noted that country-scale estimates (from AQUASTAT) were integrated with
state-level water withdrawal estimates (from USGS NWIS). The protocol for the integration of local information
with global data sources was developed for further use in the future, to enable the integration of other regional or
national datasets as soon as they become available.
**Table 6. Water demand fields, their description, data source and applied transformations; cells with bold italics show**
**required intermediate fields; name in brackets in italics next to each field correspond to the name in the data repository.**

| Field type | Description | Data source | Transformations (in order) |
|---|---|---|---|
| ***Population density*** (*pop*) | Number of people per grid cell | GHS-POP R2019A (1975, 1990, 2000, 2015) | Reprojecting and upscaling from native (9 arc sec) to the final grid and intermediate resolution of 0.01°x0.01° with sum (in total four fields); Transforming from population number to density per grid cell (i.e. dividing by grid cell area) and upscaling from intermediate to final resolution with mean (in total four fields); NoData filling (year) with linear interpolation till 2015, and with years 2000 and 2015 trend extrapolation 2016 onwards ($pop_{year}^{grid}$; in total 41 fields) |
| | | TM 'country borders', US CB 'state borders' | Shapefile (country, US State) gridding to final grid and intermediate resolution of 0.01°x0.01°, then to final resolution; Transforming from population density per grid cell to population per country (i.e. multiplying by grid cell area and summing grid cells according to the country mask from step above; $pop_{year}^{country}$; in total one table) |
| Water demand for domestic use (*dom*) | Daily supply of water volume for indoor and outdoor household purposes and for all the uses that are connected to the municipal system (e.g., water used by shops, schools, and public buildings) | AQUASTAT (per country), USGS NWIS (per US State), *pop* | Unit conversion from native to km³/year; NoData filling (year): for countries – with linear interpolation and forward/ backward extrapolation based on $pop_{year}^{country}$, for US states – with linear interpolation and nearest neighbour extrapolation ($demand_{year}^{country}$, in total one table) |
| | | *pop*, TM 'country borders', US CB 'state borders' | Transforming water demand ($demand_{year}^{country}$) to water demand per capita per country/ US State per year (in total one table): $perCapitaDemand_{year}^{country} = \frac{demand_{year}^{country}}{pop_{year}^{country}}$; NoData filling (country) with nearest neighbour; Transforming from water demand per capita to water demand per grid cell (i.e. weighting by $pop_{year}^{grid}$; in total one field per year): $demand_{year}^{grid} = perCapitaDemand_{year}^{country} \cdot pop_{year}^{grid}$ |
| | | MSWX, Huang et al. (2018) [Table 3, Eq. (2)]. | Temporal downscaling (month) to account for the withdrawal fluctuations between the warmest and coldest months based on Huang et al. (2018) Eq. (2) (in total 12 fields per year): $demand_{month,year}^{grid} = \frac{demand_{year}^{grid}}{month_{year}^{number}} \cdot \left( \frac{\overline{T}_{month,year}^{grid} - {}^{avg}\overline{T}_{year}^{grid}}{{}^{max}\overline{T}_{year}^{grid} - {}^{min}\overline{T}_{year}^{grid}} \cdot R + 1 \right)$, where ${}^{avg}\overline{T}_{year}^{grid}$, ${}^{max}\overline{T}_{year}^{grid}$, ${}^{min}\overline{T}_{year}^{grid}$ are the average, maximum, minimum monthly temperatures in a year; $\overline{T}_{month,year}^{grid}$ is the average temperature in a month of the year; $R$ is the amplitude of the monthly fluctuations from Huang et al. |

| | | | (2018) [Table 3]; $month_{year}^{number}$ is number of months in a year, i.e. 12; Temporal downscaling (day; in total 12 fields per year): $demand_{day,month,year}^{grid} = \frac{demand_{month,year}^{grid}}{day_{month}^{number}}$, where $day_{month}^{number}$ is number of days in a month of a certain year |
|---|---|---|---|
| Water demand for industrial use (*ind*) | Daily supply of water volume for fabricating, processing, washing and sanitation, cooling or transporting a product, incorporating water into a product | AQUASTAT (per country), USGS NWIS (per US State), GCAM (per region), Vassolo and Doll (2005), World Bank (MVA), *pop*, TM 'country borders' | Unit conversion from native to km³/year; NoData filling (year; in total one table): <br> • regional data – downscaling (spatial) to country values (i.e. weighting by $pop_{year}^{country}$), then linear interpolation (between years) and nearest neighbour extrapolation in time, finally rescaling values according to Vassolo and Doll (2005); <br> • country data – with linear interpolation (between years) and forward/ backward extrapolation based on *MVA* or $pop_{year}^{country}$, value disaggregation from industrial water demand to manufacturing and thermoelectric water demands according to regional data results; <br> • for US States data – with linear interpolation (between years) and nearest neighbour extrapolation; <br> • mosaicking results from US States and country data, from regional data, if not – with zero |
| | | *pop*, TM 'country borders', US CB 'state borders' | Transforming from water demand per country/ US State to per grid cell (i.e. weighting by $pop_{year}^{grid}/pop_{year}^{country}$; in total one field per year): $demand_{year}^{grid} = \frac{demand_{year}^{country}}{pop_{year}^{country}} \cdot pop_{year}^{grid}$; <br> Temporal downscaling (day; in total one field per year): $demand_{day,year}^{grid} = \frac{demand_{year}^{grid}}{day_{year}^{number}}$, where $day_{year}^{number}$ is number of days in a year |
| Water demand for thermoelectric use (*ene*) | Daily supply of water volume for the cooling of thermoelectric and nuclear power plants | AQUASTAT (per country), USGS NWIS (per US State), GCAM (per region), Vassolo and Doll (2005), World Bank (MVA), *pop*, TM 'country borders' | Same steps as for water demand for industrial use, but using the energy withdrawals as input data (in total one table) |
| | | *pop*, TM 'country borders', US CB 'state borders' | Same steps as for water demand for industrial use (in total one field per year) |
| | | GCAM (per region), MSWX, Huang et al. (2018) [Eq. (3)-(10)]. | Temporal downscaling (month) to account for the withdrawal fluctuations between the warmest and coldest months based on Huang et al. (2018) Eq. (3)-(10) (in total 12 fields per year) |
| Water demand for livestock use (*liv*) | Daily supply of water volume for domestic animal needs | AQUASTAT (per country), USGS NWIS (per US State), GCAM (per region), GLW3, TM 'country borders' | Unit conversion from native to km³/year; NoData filling (year; in total one table): <br> • regional data – spatial downscaling from regional withdrawals to country values (i.e. weighting by total livestock mass estimates per country from GLW3, $livestock_{year}^{country}$): $demand_{year}^{country} = \frac{withdrawal_{year}^{region}}{livestock_{year}^{region}} \cdot livestock_{year}^{country}$, then value linear interpolation (between years) and nearest neighbour extrapolation, finally rescaled with country data (if available) <br> • for US States data – with linear interpolation (between years) and nearest neighbour extrapolation; <br> • mosaicking results from US States and regional data, if not – with zero |
| | | GLW3, TM 'country borders', US CB 'state borders' | Transforming from water demand per country/ US State to per grid cell (i.e. weighting by $\frac{livestockDensity_{year}^{grid}}{livestockDensity_{year}^{country}}$; in total one field per year): |

| | | | $$demand_{year}^{grid} = \frac{demand_{year}^{country}}{livestockDensity_{year}^{country}} \cdot livestockDensity_{year}^{grid};$$ Temporal downscaling (day; in total one field per year): $$demand_{day,year}^{grid} = \frac{demand_{year}^{grid}}{day_{year}^{number}},$$ where $day_{year}^{number}$ is number of days in a year |
|---|---|---|---|

To the best of the authors' knowledge, no other publicly accessible temporally varying global water demand field set exists (only static datasets). A rigorous validation of the temporally varying water demand fields is not straightforward at the global scale, as the only comprehensive global data source, FAO AQUASTAT, was used to create the fields.

## 8.3 Regional examples

In general fields in water demand category are easy to interpret as they show how much water per day is needed to satisfy certain type of human induced needs. In reality water demand fields are mainly covering urbanised areas and are scattered around (i.e. not continuously looking field), with relatively small variations in field values from month to month. Example for domestic water use is presented for August 2018 in Figure 19 for Po River area in 1 arc min and 3 arc min resolution, and in Figure 20 for Amazon River and Brahmaputra River areas at 3 arc min resolution.

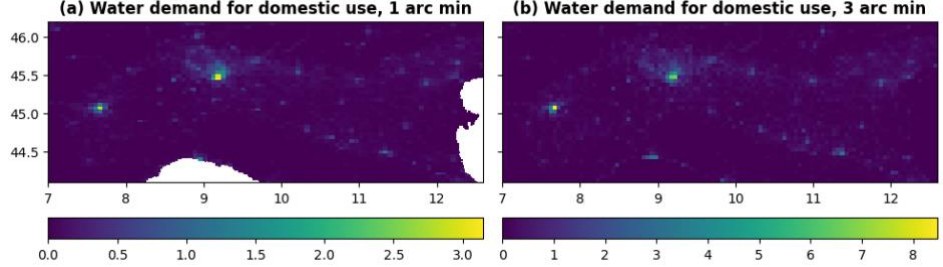

**Figure 19. Water demand for domestic use in mm per day at 1 arc min (~1.9 km at the equator, left column, plot a) and 3 arc min (~5.6 km at the equator, right column, plot b) resolution for Po River area in Italy.**

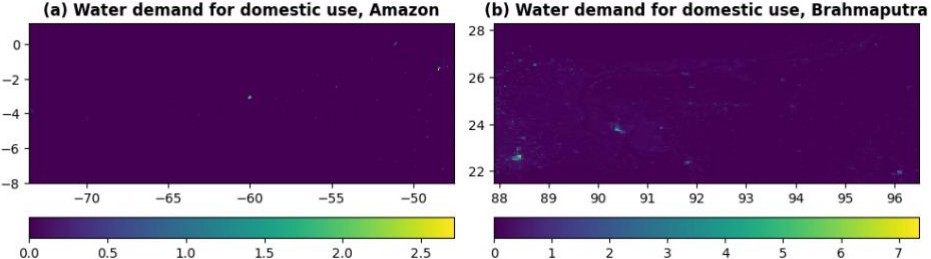

**Figure 20. Water demand for domestic use in mm per day at 3 arc min (~5.6 km at the equator) resolution for Amazon River area (left column, plot a) and Brahmaputra River area (right column, plot b).**

## 9 Data, access, licensing, documentation

The new CEMS_SurfaceFields_2022 is an open-source dataset of the Copernicus Emergency Management Service describing key components of the Earth surface generally required in environmental and hydrological modelling, including Earth system modelling and numerical weather prediction. The dataset includes static fields (e.g. forest fraction), yearly cycle fields (e.g. 10-day average LAI, in total 36 fields), and yearly varying fields (e.g. water demand). The surface fields are based on 25 different sources, including global and regional high resolution (up to 100 m) gridded and vector datasets. They were processed into two set of fields (i) at 1 arc min resolution (~1.86 km at the equator) over Europe (72.25 N/ 22.75 N, 25.25 W/ 50.25 E; 4530x2970 grid cells), and (ii) at 3 arc min resolution (~5.57 km at the equator) over the Globe (90.00 N/ 90.00 S, 180.00 W/ 180.00 E; 7200x3600 grid cells), to provide an up-to-date surface state for six main field groups: (1) catchment morphology and river network, (2) land use fields, (3) vegetation properties, (4) soil properties, (5) lakes, (6) water demand. The CEMS_SurfaceFields_2022 dataset consist in total of 140 gridded fields at EPSG:4326 – WGS84: World Geodetic System projection in NetCDF format with information on Earth's surface state (see Table 9 for the full list of fields), which are grouped thematically in sub-folders. The 1 arc min European fields have a total volume

of 9.3 GB and the 3 arc min global fields have a total volume of 22.7 GB. The CEMS_SurfaceFields_2022 dataset is freely available for download from the JRC Data Catalogue (https://data.jrc.ec.europa.eu/). The set of global surface fields at 3 arc min resolution can be found here (JRC Data Catalogue – LISFLOOD static and parameter maps for GloFAS – European Commission (europa.eu), https://data.jrc.ec.europa.eu/dataset/68050d73-9c06-499c-a441-dc5053cb0c86) and the set of surface fields for the European domain at 1 arc min resolution can be found here (JRC Data Catalogue – LISFLOOD static and parameter maps for Europe – European Commission (europa.eu), https://data.jrc.ec.europa.eu/dataset/f572c443-7466-4adf-87aa-c0847a169f23). The README.txt file that can be found there contains the basic description of each surface fields including general information, data description, file overview, methodological information and data access and sharing information (for detailed technical description of how the surface fields were generated refer to the LISFLOOD User Guide, available online: https://ec-jrc.github.io/lisflood-code/4_Static-Maps-introduction/). The changelog.txt file – provides users with information on updates to the datasets. The copyright.txt file – information about the data license (CC BY 4.0).

**Table 9. Full list of surface fields with short description and units included in CEMS_SurfaceFields_2022 dataset; name in italics correspond to the field's file name in the data repository.**

| Field group | Description | Name | Units |
|---|---|---|---|
| Main | model's field (i.e. in technical for model operation/ running sense) | *mask* | dimensionless |
| Catchment morphology and river network | local drainage direction (i.e. flow direction from one cell to another) | *LDD* | dimensionless |
| | grid cell area | *pixarea* | $m^2$ |
| | grid cell length | *pixlength* | m |
| | upstream drainage area | *upArea* | $m^2$ |
| | standard deviation of elevation | *elvstd* | m |
| | gradient | *gradient* | m/m |
| | channel bottom width | *chanbw* | m |
| | channel length | *chanlenght* | m |
| | channel gradient | *changrad* | m/m |
| | Manning's roughness coefficient for channels | *chanman* | $s/m^{1/3}$ |
| | channel mask (i.e. presence of river channel) | *chan* | dimensionless |
| | channel side slope (i.e. channel's horizontal distance divided by vertical distance) | *chans* | m/m |
| | bankfull channel depth | *chanbnkf* | m |
| | channel floodplain (i.e. width of the area where the surplus of water is distributed when the water level in the channel exceed the channel depth) | *chanflpn* | m |
| Land use fields | fraction of forest | *fracforest* | dimensionless |
| | fraction of sealed surface | *fracsealed* | dimensionless |
| | fraction of inland water | *fracwater* | dimensionless |
| | fraction of irrigated crops | *fracirrigated* | dimensionless |
| | fraction of rice | *fracrice* | dimensionless |
| | fraction of other cover types | *fracother* | dimensionless |
| Vegetation properties (for forest [f], irrigated crops [i], other land cover types [o]) | crop coefficient | *cropcoef_f, cropcoef_i, cropcoef_o* | dimensionless |
| | crop group number | *cropgrpn_f, cropgrpn_i, cropgrpn_o* | dimensionless |
| | Manning's surface roughness coefficient | *mannings_f, mannings_o,* | $s/m^{1/3}$ |
| | rice planting days (3 seasons) | *riceplantingday1, riceplantingday2, riceplantingday3* | calendar day number |
| | rice harvesting days (3 seasons) | *riceharvestday1, riceharvestday2, riceharvestday3* | calendar day number |
| | leaf area index | *laif, laii, laio* | $m^2/m^2$ |
| Soil properties (for [1, 2, 3] layers; for forest [f], non-forest [o]) | surface layer depth | *soildepth1_f, soildepth1_o* | mm |
| | middle layer depth | *soildepth2_f, soildepth2_o,* | mm |
| | subsoil depth | *soildepth3_f, soildepth3_o* | mm |
| | saturated volumetric soil moisture content | *thetas1_f, thetas1_o, thetas2_f, thetas2_o, thetas3* | $m^3/m^3$ |
| | residual volumetric soil moisture content | *thetar1, thetar2, thetar3* | $m^3/m^3$ |
| | pore size index | *lambda1_f, lambda1_o, lambda2_f, lambda2_o, lambda3* | dimensionless |
| | Van Genuchten equation parameter | *genua1_f, genua1_o, genua2_f, genua2_o, genua3* | $cm^{-1}$ |

| | saturated soil conductivity | *ksat1_f, ksat1_o, ksat2_f, ksat2_o, ksat3* | mm/day |
|---|---|---|---|
| Lakes | lake mask (i.e. presence of lakes) | *lakemask* | dimensionless |
| Water demand | livestock | *liv* | mm/day |
| | industry | *ind* | mm/day |
| | thermoelectric production | *ene* | mm/day |
| | domestic use | *dom* | mm/day |


Whilst the CEMS_SurfaceFields_2022 dataset followed strict requirements of the LISFLOOD-OS model (e.g.
format, treatment of missing values, number of soil layers, etc…) it definitely can be used outside the LISFLOOD
context, using the full dataset or its parts, for applications such as modelling risk assessment. The workflow and
methodology used to generate the dataset and published in this manuscript can be used as reference and be easily
modified if further adaptation to the dataset is needed (e.g. using different set of equations to describe the soil
properties, or sourcing new/ more relevant local datasets).
**10      Conclusion**
The Earth's surface has a strong impact on the surface energy and water balance that drives lower atmosphere
weather conditions and river discharge fluctuations. Depending on the surface type (e.g. land use, terrain or soil),
weather in the region can be colder/ warmer, more/ less humid, drier/ rainier, and/ or calmer/ windier than its
surroundings, and the terrestrial water cycle can differ, with water infiltrating more/ less in the soil, leaving as
evaporation in a larger/ smaller rate, and reaching rivers faster/ slower. Surface information is provided by land
use and ecosystem type (e.g., forest, rice paddy, bare ground, urban), river geometry (e.g., channel width, channel
length), soil properties (e.g., depth, porosity, hydraulic properties), amongst others.
Information of underlying surface fields can be accounted for in Earth system and environmental models (e.g.
atmospheric, hydrological, etc.) to simulate the evolution in space and time of water, energy and carbon cycles. If
artificial influences and human intervention are included within the modelled processes (e.g. irrigation or water
management through reservoirs), the information required to describe the processes must also be integrated within
the modelling framework. Generally, this is achieved through a set of independent files used as input to the models.
Because of the temporal non-stationarity of some surface fields, typically associated with human intervention such
as land use and water use, but also due to climatic variation such as lake extent (new lakes forming or lakes
shrinking), input surface fields must be as representative as possible to the simulated period of interest. For
medium-range forecasting systems, this should be as close from present as possible, for example. When simulating
long periods, especially looking at past or future decades, caution must be given to results especially if some
surface fields which have substantially changed during the simulation period do not explicitly incorporate time
and instead are based on the most recent period, as they may not be representative to the full study period and can
introduce substantial biases that grow with time. Same is applicable if surface fields are used for collecting
statistical data in general, as stats based on stationary fields represent only the period used to generate stationary
field in question.
In addition, in recent years the horizontal resolution of global Earth system and environmental models has been
constantly increasing reaching the kilometre scale milestone, supported by the technological developments in the
field of High Performance Computers and the wealth of high resolution datasets freely available. This imposes
another condition to the input surface fields – it has to be of rather high horizontal resolution (i.e. ~2 and 6 km at
the equator).
Thanks to the availability of a wide range of high resolution environmental data derived from the use of ground,
unconventional and satellite measurement sensors, new high resolution datasets describing the Earth's surface are
nowadays released regularly. Even though each dataset may have a very low absolute and root mean square errors
compared against available independent data, merging different datasets for modelling purposes (e.g. to model
hydrological surface parameters) might lead to questionable results and even model crash, due to possible
discontinuity or inconsistency in the combined datasets. In the specific case of hydrological modelling where river
flow is also represented, high horizontal resolution does not guarantee better modelling per se. Sources of
potentially large errors can be easily hidden in high resolution datasets. This is the case for instance of errors in
the Digital Elevation Models when they are used to obtain the rivers drainage network. Small errors in the
elevation of a grid cell can lead to a totally inaccurate representation of the location and the direction in which the
river is flowing in the model compared to reality. Mislocating a river or having a slightly inaccurate catchment
area can represent a trivial inaccuracy for most applications, but it can also lead to missed flood warning for
thousands of people within a flood awareness system. To benefit from different recent high resolution datasets
based on satellite and ground measurements, it is essential that a well-defined, thorough workflow is designed and
implemented so that the final products are consistent and compatible with each other, and can be used in
combination.
The work presented in this manuscript is focused not only on the final surface field set generation (i.e.
CEMS_SurfaceFields_2022), but also on deriving robust reproducible methodology that could be re-applied once
new versions of 25 or less input sources are released. Understanding of the methodology applied helps to interpret
values in the final surface fields and possibly even numerical model results that use these surface fields. The
collection of input sources and their preparation for actual use is a very important step as it includes going through
all technical documentation, comparison and verification of papers, and investigation of the actual data, as well
as data gridding, interpolation, and scaling. All input sources for CEMS_SurfaceFields_2022 are ranked according
to their quality and up-to-date in order to favour one value in ambiguous situations when several datasets provide
different information for the same location. Consistency check between all surface type fractions is carried out to
address that ambiguity during the merge of information of different origin (i.e. adjust fractions to sum to one in
each grid cell). Some fields, like forest fraction, were rather straightforward to create from available source, yet it
was noted that prior correction of the source was needed to delete erroneous forest grid cells from the Fox Basin
in Canada (the mismatch was only spotted during the investigation of the actual data, as it was absent from the
documentation). Other fields, like soil hydraulic properties, are created not only from the source information but
also from the forest fraction that had to be generated prior; the soil hydraulic property methodology also includes
several steps that have to be performed at the data native resolution (i.e. 250 m) using information from several
global fields simultaneously which becomes technically and computationally challenging. Surface fields with
clear multi-annual changes, like water demand maps, are created using temporal interpolation and extrapolation
from multiple data sources to create time series fields. A final and non-trivial task is to have all resulting fields on
the identical required grid without deterioration of the actual value precision, even after several file type
translations (e.g. local drainage direction field can be automatically checked and corrected if needed for required
boundaries only in PCRaster format, not NetCDF). Due to the number of data sources and surface fields required
to represent the main variables (i.e. 70) used in Earth system and environmental models, the overall effort to
generate the CEMS_SurfaceFields_2022 dataset (both human and computing resources) was substantial.
The CEMS_SurfaceFields_2022 dataset is a new data source open to all offering a kilometre-scale resolution of
high-quality data describing the Earth's surface (openly available online from the data catalogue of the JRC for
Europe at ~1.9 km at the equator or 1 arc min resolution: https://data.jrc.ec.europa.eu/dataset/f572c443-7466-
4adf-87aa-c0847a169f23, and for Globe at ~5.6 km at the equator or 3 arc min resolution:
https://data.jrc.ec.europa.eu/dataset/68050d73-9c06-499c-a441-dc5053cb0c86, last accessed: 21.01.2024),
providing exceptional opportunity for the research and scientific community to extend and multiply European and
global applications in wide ranging fields of the water-energy-food nexus. The CEMS_SurfaceFields_2022
surface fields use can be vast, e.g. standard deviation of elevation and other orographic sub-grid parameters are
critical for radiation parametrization, especially for shadowing effect; channel geometry fields are vital to describe
overbank inundation and infer inundated areas in wetland methane and soil carbon modelling; land use fractions
are needed for skin temperature calculations, biogenic flux calculations, urban planning, and climate mitigation
plan preparation; LAI use include biomass allocation, which can be used for fire danger forecasting, and carbon
stock monitoring, whilst rice planting/ harvesting days are important for yearly cycle of methane modelling; soil
properties are used for soil moisture calculations; and the area covered by lakes can be used for computing
evaporation from open water, freshwater storage, unregulated surface water extent, fresh water scarcity indexes,
and biogenic green house gas emission, as well as for reproducing different climate mitigation scenarios. All of
the above state that CEMS_SurfaceFields_2022 surface fields can be used for weather prediction, Earth system
modelling, hydrological and environmental modelling, or statistical analysis in general, with a spatial scale
allowing for global, regional and even national applications.

*Data availability.* The CEMS_SurfaceFields_2022 datasets are freely available for download from the JRC Data
Catalogue – global at ~5.6 km at the equator or 3 arc min resolution:
https://data.jrc.ec.europa.eu/dataset/68050d73-9c06-499c-a441-dc5053cb0c86), over Europe at ~1.9 km at the
equator or 1 arc min resolution: https://data.jrc.ec.europa.eu/dataset/f572c443-7466-4adf-87aa-c0847a169f23,
and are documented in this paper.
*Author contributions.* CP and PS shaped initial plan of the research; MC and FM executed initial plan; CM, SG
and JD reviewed initial results and provided guidance in further research. MC, FM and CP prepared a first draft
of the paper, which was adapted to its present state by contributions from CM, SG, JD, PS and HB.
*Competing interests.* The authors declare that they have no conflict of interest.
*Acknowledgements.* CEMS_SurfaceFields_2022 is a product and service of the Copernicus Emergency
Management Service. Financial support for MC, FM, CM and CP was provided by contract 941462-IPR-2021.

Authors thank two anonymous reviewers for their valuable comments and suggestions that helped to shape the manuscript to it's current state.

*Financial support.* This research has been supported by contract 941462-IPR-2021.

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

# Appendix

## Appendix 1

All data sources used to produce dataset's surface fields, mentioned in Sections 3 to 9, are described here. All data considered were open source, freely available, updated as recently as possible, with recognised reference on their quality.

### 1.1 Catchment morphology and river network

**The MERIT DEM: Multi-Error-Removed Improved-Terrain Digital Elevation Model v.1.0.3** [15 October, 2018] (further referred as MERIT DEM) is a high accuracy global DEM at 3 arc second resolution (~90 m at the equator) covering land area from 90 N to 60 S, selected for its ability to clearly represent landscapes such as river networks and hill-valley structures even in flat areas where height errors could be larger than topography variability (Yamazaki et al., 2017; Bhardwaj, 2021; Chai et al., 2022). It is derived from seven different open-source datasets, delivered as 57 GeoTiff files 30º by 30º region each, at ~90 m resolution (in total 90.0 GB), representative of the year 2018. More detail on method, data content and access can be found in Yamazaki et al. (2017) and MERIT DEM web-page http://hydro.iis.u-tokyo.ac.jp/~yamadai/MERIT_DEM.
The MERIT DEM was used to compute standard deviation of elevation, gradient and channel geometry fields.

**The Catchment-based Macro-scale Floodplain (CaMa-Flood) Global River Hydrodynamics Model v4.0 maps** (further referred as CaMa-Flood) are used for the basic maps describing all physical properties of the river network. It is derived from MERIT Hydro (MERIT Hydro is a global hydrography dataset, created by using elevation (i.e. MERIT DEM) and several inland water maps; more detail can be found in Yamazaki et al. (2019) and MERIT Hydro web-page http://hydro.iis.u-tokyo.ac.jp/~yamadai/MERIT_Hydro) for high resolution river routing applications using the FLOW algorithm (Yamazaki et al., 2009; Yamazaki et al., 2011). The maps include information on channel length, river topography parameters, floodplain elevation profile, channel width and channel depth. The maps exist at 15, 6, 5, 3 and 1 arc min resolutions covering land area from 90 N to 60 S, representative of the year 2017, and for each resolution, they are available as one single file with all variables in NetCDF format (for 1 arc min 737.0 MB). More detail on method, data content and access can be found in Yamazaki et al. (2011) and CaMa-Flood web-page http://hydro.iis.u-tokyo.ac.jp/~yamadai/cama-flood/index.html. Note that whilst the CaMaFlood maps where originally generated for the specific use of the CaMa-Flood model, they can also serve as basic to derive alternative maps for other environmental models, as done here.
The CaMa-Flood maps were used to create the local drainage direction (LDD), upstream drainage area, channel geometry and land masks fields.

### 1.2 Land use fields

**The Copernicus Global Land Service (CGLS) Land Cover (LC) 100m map** (further referred as CGLS-LC100) is a global land cover map of the year 2015 (Buchhorn et al., 2020). It is derived from the PROBA-V 100 m satellite image collection, a database of high quality land cover training sites and ancillary datasets, reaching an accuracy of 80 % at Level1 (Buchhorn et al., 2021). It contains 23 classes for discrete classification and 10 classes for continuous cover fractions; and it is delivered as 15 files in GeoTiff format (in total 39.3 GB) at 100 m resolution covering land area from 90 N to 60 S and representative of the year 2015. More detail on method, data content and access can be found in Buchhorn et al. (2021) and Copernicus web-site https://land.copernicus.eu/global/products/lc.
The CGLS-LC100 was used to generate crop parameters and Manning's surface roughness coefficient for forest and other land cover types, to generate forest, inland water, and sealed surface fraction fields, following a basic quality check on large water bodies (i.e. correcting Fox Basin and Caspian Sea).

**The Coordination of Information on the Environment (CORINE) Land Cover (CLC) inventory for 2018** (further referred as CLC2018) is a set of maps describing the land cover/ land use status of 2018 covering

39 countries in Europe with a total area of over 5.8 Mkm$^2$. The dataset is derived from satellite imagery (mainly
Sentinel-2, based on a constellation of two satellites orbiting Earth at altitude of 786 km 180° apart revisiting
equator every 5 days, and for gap filling Landsat-8, making a constellation together with Landsat-9 satellite
orbiting Earth at altitude of 705 km each revisiting equator every 16 days) and in-situ data and contains 44 classes,
delivered as one GeoTiff raster file (125.0 MB) at 100m resolution covering land area over Europe, representative
of the time period 2017-2018. The overall accuracy for CLC2018 is 92 % for the blind analysis (i.e. validation
team had no knowledge of the CLC2018 thematic classes) but there are regional variations: the Black Sea
geographical region has the lowest accuracy of 84 %; country-wise overall accuracy vary from 86 % for Portugal
to 99 % for Iceland, lowest accuracy being linked to the landscape complexity (Moiret-Guigand, 2021). More
detail on method, data content and access can be found in Büttner and Kosztra (2017) and Moiret-Guigand (2021),
and Copernicus web-site https://land.copernicus.eu/pan-european/corine-land-cover/clc2018.
The CLC2018 was used to generate the irrigated crop fraction and rice fraction fields.
**The Spatial Production Allocation Model (SPAM) – Global Spatially-Disaggregated Crop Production**
**Statistics Data for 2010 v2.0** (further referred as SPAM2010) is a global dataset generated in 2020, which
redistributes crop production information from country and sub-national provinces level to a finer grid cell level
(IFPRI, 2019). It is derived from numerous data sources, including crop production statistics, cropland data,
biophysical crop "suitability" assessments, spatial distribution of specific crops or crop systems, and population
density. SPAM2010 contains estimates of crop distributions within disaggregated units (based on a cross-entropy
approach) for 42 crops and two production systems (irrigated and rainfed), and is delivered as 84 files in shapefile
format at 10 km (5 arc min) resolution covering land area from 90 N to 60 S and representative of the year 2010
(in total 2.2 GB). Based on crop expert judgement from international (i.e. International Rice Research Institute,
International Maize and Wheat Improvement Center) and national organisations (i.e. The Chinese Academy of
Agricultural Sciences) SPAM2010 over Europe and America is more accurate than over Africa and South East
Asia, with best performance in allocating rice; grid-by-grid comparison of crop areas with independent Cropland
Data Layer (produced by using satellite images and vast amount of ground truth) over continental United States
shows coefficient of determination (R$^2$) 0.7-0.9 and root mean square error (RMSE) 231-307 ha indicating a
relatively high reliability, with highest R$^2$ and lowest RMSE values are for maize and soybean (Yu et al., 2020).
More detail on method, data content and access can be found in Yu et al. (2020) and MapSPAM web-site
https://mapspam.info.
SPAM2010 was used to compute the irrigated crop and rice fractions, crop parameters and Manning's surface
roughness coefficient for irrigated crop fields.
**1.3 Vegetation properties**
**The Food and Agriculture Organisation (FAO) of the United Nations Irrigation and Drainage Paper No.**
**56** (further referred as FAO56) is a publication covering geographically referenced statistics for crop development
stages, crop coefficients, crop height, rooting depth, and soil water depletion fraction for common crops found
across the world; it also covers procedures for information aggregation, e.g. on the grid. It is delivered as an article
with a set of tables and equations and can be considered as the most complete source of information on crop
properties. More detail on method and data content can be found in Allen et al. (1998) and FAO online crop
information web-page http://www.fao.org/land-water/databases-and-software/crop-information/tobacco/en/.
FAO56 was used to compute the crop coefficients for forest, irrigated crops and other land cover types (online
crop information was specifically used for tobacco); and for intermediate computations such as depletion fraction
for different crop and surface types (table), crop height and root depth fields.
**Intara** et al. (2018) is a publication covering oil palm roots architecture.
Intara et al. (2018) was used for oil palm root depth information in addition to FAO56.
**Burek** et al. (2014) is a publication covering summarised information for crop coefficients, rooting depth, crop
group number and Manning's surface roughness coefficient for different surface types.
Burek et al. (2014) was used for built-up, bare/ sparse vegetation, snow & ice, permanent inland water, ocean &
seas, herbaceous wetland, moss & lichen surface types crop coefficients, rooting depth, crop group number and
Manning's surface roughness coefficient information in addition to FAO56 and other sources.
**The Wofost 6.0 crop simulation model description** (further referred as SUPIT) is a publication on developing,
validating, and testing new or already existing agrometeorological models (Supit et al., 1994). It contains crop
group information for several crops as examples, and relation of a crop group from water depletion fraction. The
publication is delivered as a book with a set of tables and equations. Information on crop group is still considered
up-to-date. More detail on method and data content can be found in Supit et al. (1994).
SUPIT was used to compute the crop group fields for forest, irrigated crops and other land cover types.
**The Open-Channel Hydraulics manual** (further referred as CHOW) is a publication on open-channel
hydraulics, including basic principles and different types of flows, i.e. uniform, gradually varied, rapidly varied,
and unsteady (Te Chow, 1959). It contains information on roughness coefficient over different surfaces. The
publication is delivered as a book with a set of tables and equations. More detail on method and data content can
be found in Te Chow (1959).
CHOW was used to compute the Manning's surface roughness coefficient fields for forest, irrigated crops and
other land cover types.
**The Copernicus Global Land Service (CGLS) Leaf Area Index (LAI) 1km Version 2 collection** (further
referred as CGLS-LAI) is a set of global maps without missing data describing vegetation dynamics – the annual
evolution of LAI at 10-day intervals over the period of 1999-2020. The dataset is derived from
SPOT/VEGETATION and PROBA-V data. The dataset's root mean square deviations over 20 GBOV sites over
the period 2014-2018 is 0.92, compared to 1.19 for MODIS C6 LAI product (Martinez-Sanchez, 2020). The
dataset is delivered as one multi-band file per year in NetCDF (netCDF4 CF-1.6) format (14.7 GB per year) at 1
km resolution covering land area from 90 N to 60 S and representative of the 10-year period of 2010-2019. More
detail on method, data content and access can be found in Smets (2019) and Martinez-Sanchez (2020), and
Copernicus web-site https://land.copernicus.eu/global/products/lai.
CGLS-LAI was used to compute the LAI fields for forest, irrigated crops and other land cover types.
**The RiceAtlas v3** (further referred as RiceAtlas) is a spatial database of global rice calendars and production. It
contains information on start, peak and end dates of sowing, transporting and harvesting rice, derived from global
and regional databases, national publications, online reports, and expert knowledge. It is delivered as 7 files in
shapefile format (in total 195.8 MB) for administrative units (in total 2725 spatial units) at 1 km resolution for the
national production totals to match the years 2010-2012 (Laborte et al., 2017a). RiceAtlas is ~10 times more
spatially detailed, and has ~7 times more special units comparing with other global datasets (Laborte et al., 2017b).
More detail on method, data content and access can be found in Laborte et al. (2017a) and Laborte et al. (2017b).
RiceAtlas was used to compute rice planting and rice harvesting days for three different seasons.
**1.4 Soil properties**
**The International Soil Reference and Information Centre (ISRIC) SoilGrids250m global gridded soil**
**information release 2017** (further referred as SoilGrids250m) is an output of special predictions produced by the
SoilGrids system, as a set of global soil property and class maps at 250 m resolution. It is derived from soil profile
data (from ~150,000 sites globally) with the use of machine learning, and contains information on soil
characteristics at six standard depths, including soil textures (clay, silt, sand), depth to bedrock, bulk density,
organic carbon, pH and cation exchange capacity. It is delivered as 43 files in GeoTiff format (in total 111.8 GB)
at 250 meters resolution covering land area with no permanent ice and representative for the year 2010 (according
to land cover) (Hengl et al., 2017). SoilGrids250m pH comparison with SSURGO data over California (depth 0-
200 cm) and Soil and Landscape Grid of Australia data over Tasmania (depth 0-5 cm) show high correlation, 0.79
and 0.71 respectively (Hengl et al., 2017). Despite its limited accuracy (i.e. between 30 and 70 %, according to
the SoilGrids web-site) due to the scarcity of soil profile observations (especially in Central Asia, Artic regions
costal area and desert), low resolution of covariates data and algorithms, it was selected as the most recent source
of information. More detail on method, data content and access can be found in Hengl et al. (2017) and
SoilGrids250m web-site https://www.isric.org/explore/soilgrids/faq-soilgrids-2017.
SoilGrids250m was used to compute the soil depth and soil hydraulic properties for forest and non-forest.
**1.5 Lakes**
**The Global Lakes and Wetlands Database** (further referred as GLWD) is a global database of water bodies. It
is derived from a combination of global and regional lake data sets, registers and inventories (i.e. point information
with descriptive attributes), and digital maps (i.e. polygons, rasterised global land cover and land use maps). The
database consists of two global files in shapefile format at spatial resolutions of up to 1:1 million – GLWD-1 with
3067 largest lake and 654 largest reservoir polygons (6.4 MB), and GLWD-2 with ~250000 smaller lake and
reservoir polygons (32.0 MB); and of one global file in ADF raster format at 30 arc sec resolution – GLWD-3
combines GLWD-1, GLWD-2 and additional information (8.9 MB). Validation against documented data shows
that GLWD represents good wetland maximum extent, and describes comprehensively lakes with surface area
greater or equal 1 km² (Lehner and Döll, 2004). More detail on method, data content and access can be found in
Lehner and Döll (2004) and GLWD web-site https://www.worldwildlife.org/pages/global-lakes-and-wetlands-
database.
GLWD (i.e. only GLWD-1 and GLWD-2) was used to compute the discrete lake mask field.

**1.6 Water demand**

**AQUASTAT** is the FAO's global information system on water resources and agricultural water management.
AQUASTAT collects information on water use via the network of AQUASTAT National Correspondents who
are required to fill the annual questionnaire and collaborate with AQUASTAT team in the data validation process.
Five types of manual checks are followed by automatic implementation of almost 200 validation rules. The dataset
includes data for 180 countries worldwide, yearly data from 1979 to 2019 were used to produce the maps presented
by this manuscript. Float, lumped values for each country for the variables "Gross Domestic Product (GDP)",
"Industry, value added to GDP", "Agricultural water withdrawal", "Industrial water withdrawal", "Municipal
water withdrawal", "Total water withdrawal", and "Irrigation water withdrawal" were obtained in CSV format (2
files, in total 2.0 MB) from the AQUASTAT data acquisition dashboard
(https://tableau.apps.fao.org/views/ReviewDashboard-v1/country_dashboard). More detail on method, data
content and access can be found in AQUASTAT web-site
https://www.fao.org/aquastat/en/overview/methodology/.
AQUASTAT variables were used accordingly to compute water demand fields for domestic, industrial, energy,
livestock use.
**United States Geological Survey National Water Information System** (further referred as USGS NWIS) is a
national database on water use data for the United States (US) with annual statistics provided every 5 years since
1950. The water use data are best estimates produced by the USGS in cooperation with local, state, and federal
agencies as well as academic and private organisations. The water use data are lumped values (float numbers) for
each state, delivered in plain text format (52 files, in total 56.0 MB). Following variables were used: "Domestic
total self-supplied withdrawals, fresh, in Mgal/d", "Public Supply total self-supplied withdrawals, fresh, in
Mgal/d", "Industrial total self-supplied withdrawals, fresh, in Mgal/d", "Total Thermoelectric Power total self-
supplied withdrawals, fresh, in Mgal/d", "Total Thermoelectric Power power generated, in gigawatt-hours", and
"Livestock total self-supplied withdrawals, fresh, in Mgal/d". More detail on method, data content and access can
be found in USGS NWIS web-site https://waterdata.usgs.gov/nv/nwis/wu. For this study, data from 1985 to 2015
were used.
USGS NWIS variables were used accordingly to refine the global water demand fields for the domestic, industrial,
energy, livestock use sectors for the US.
**Global Change Analysis Model** (further referred as GCAM) is an integrated, multi-sector model developed by
the Joint Global Change Research Institute (JGCRI) to explore the overall behaviour of human and physical
systems dynamics and interactions. GCAM includes five main systems. One of these systems, the water module,
provides information about water withdrawals for energy, agriculture, and municipal uses as lumped values of
235 hydrologic basins; a detailed explanation can be found in Calvin et al. (2019). Estimates of industrial,
thermoelectric water withdrawals (energy sector) and electricity consumption were computed by running the
GCAM model, the output used are two files in CSV format (in total 4.0 MB). Data from the following sectors was
used: "biomass", "electricity", "nuclearFuelGenII", "nuclearFuelGenIII", "regional coal", "regional natural gas",
"regional oil", "SheepGoat", "Beef", "Dairy", "Pork", and "Poultry". More detail on method, data content and
access can be found in the documentation of the open source package https://github.com/JGCRI/gcam-
core/tree/gcam-v6.0.
GCAM variables were used accordingly to estimate water withdrawals for industrial, energy, livestock use.
**Global-scale gridded estimates of thermoelectric power and manufacturing water use** (further referred as
Vassolo and Doll, 2005) is a global-scale gridded estimate of water withdrawal for cooling of thermal power
stations and for manufacturing. Estimates of values for the year 1995 are provided with a spatial resolution of 0.5°
by 0.5°. Thermoelectric power water use is based on the geographical location of 63590 thermal power stations.
Manufacturing water use is computed by estimating country-specific water withdrawal values, and spatial
downscaling using city night-time lights. Dataset verification of Vassolo and Doll (2005) showed satisfactory
representation of thermoelectric power water use but high uncertainty in the representation of manufacturing water
use. The data are delivered as one shapefile (2.5 MB). More details on method, data content and validation, and
data access can be found in Vassolo and Doll (2005).
Vassolo and Doll (2005) dataset was used for the computation of energy demand fields.
**The Gridded Livestock of the World (GLW) version3** (further referred as GLW3) is a spatial gridded dataset
of the global distribution of eight livestock species for 2010. It is delivered as 8 GeoTiff files at 0.083333° (~10
km at the equator) resolution (in total 208.0 MB). The species abundance was converted to total livestock mass.
More detail on method, data content and access can be found in Gilbert et al. (2018).
GLW3 was used to spatially disaggregate the water demand for livestock use.
**World Bank manufacturing value added and gross domestic product** (further referred as World Bank) data
provide "Manufacturing, value added (constant 2015 US$)" values (further referred as MVA) and "Gross
Domestic Product GDP (constant 2015 US$)" values. The data provided as a table, downloaded in CSV format
(6 files, in total 6.0 MB) from https://data.worldbank.org.
World Bank dataset was used to temporally downscale the values of water demand fields for the industrial and
energy sectors.
**The Global Human Settlement Population Grid multitemporal version R2019A** (further referred as GHS-
POP) is a spatial raster dataset that depicts the distribution of population, expressed as the number of people per
grid cell (Freire et al., 2016; Florczyk et al., 2019; Schiavina et al., 2019). GHS-POP residential population
estimates for target years provided by CIESIN GPWv4.10 were disaggregated from census or administrative units
to grid cells, informed by the distribution and density of built-up as mapped in the Global Human Settlement
Layer. The dataset has a spatial resolution of 9 arc sec (~300 m at the equator) resolution and is delivered as
individual files in GeoTiff format for 1975, 1990, 2000 and 2015 (4 files, in total 6.5 GB; available online:
https://ghsl.jrc.ec.europa.eu/ghs_pop2019.php, last accessed: 21.01.2024).
GHS-POP was used to spatially disaggregate the country, state, basin-level information of domestic, industrial,
energy water withdrawal.
**Thematic Mapping Country Borders** shapefile (further referred as TM 'country borders') was derived from
Thematic Mapping ™, which is a tool enabling web browsers to create thematic maps and associated world
datasets. For this work, the TM World Borders Dataset was downloaded as one shapefile (10.0 MB). **The United**
**States Census Bureau** Cartographic Boundary Files – Shapefile (further referred as US CB) provides the State
boundaries for the USA. For this work, the 2018 version was retrieved as one shapefile (3.2 MB; available online:
https://www.census.gov/geographies/mapping-files/time-series/geo/carto-boundary-file.html, last accessed:
21.01.2024). More detail on method, data content and access can be found in
http://thematicmapping.org/downloads/.
TM 'country borders' and US CB were used to spatially disaggregate the information of water withdrawal for
domestic, industrial, energy use.
**Multi-Source Weather** (further referred as MSWX) is a high-resolution (3-hourly, 0.1°), bias-corrected
meteorological product with global coverage from 1979 to 7 months into the future. The data for 42 years
(~316700 files in NetCDF format, in total 128.0 GB) were retrieved via www.gloh2o.org/mswx/. For more
detailed information, see Beck et al. (2022).
MSWX 2-meter daily and monthly maximum and minimum air temperature were used to account for the climate-
induced intra- and inter- annual fluctuations of domestic, livestock, and energetic water demand.
**Huang et al. (2018)** is a publication presenting 0.5° resolution global monthly gridded sectoral water withdrawal
dataset for the period 1971–2010.
Huang at al. (2018) Table 3 (calibrated R coefficient values) and Eq. (2) to (6) for temporal downscaling of
domestic and energy water demands were used in this study, respectively.

## Appendix 2

Unit conversion to fraction
Hectare (ha): $fraction = {ha \cdot 10^4}/{GridCellArea_{m^2}}$;
Percentage (%): $fraction = {\%}/{100}$;
Class (landcover type): $fraction = 1$, i.e. assumes full 100 % coverage of the grid cell.

## Appendix 3

Soil depth
Soil depth layers are derived following Burek et al. (2014) in which the total soil depth is horizontally divided in
three layers. The total soil depth is the 'absolute_depth_to_bedrock' from SoilGrids250m, whereas root depths of
forest and non-forest are derived from FAO56 and CGLS-LC100 dataset at SoilGrids250m native (~250 m)
resolution (see Section 6.2 for more details). The methodology implemented for the creation of three soil layers
is the following:
Soil depth layer 1 (surface) $SD_1$ is assumed constant, equal to 50 mm all over the world for consistency with
satellite-derived datasets (satellite signal penetration depth of 50 mm is a good approximation to take into account
different meteorological conditions at different hour of the day globally based on Lv et al. (2018)), and follow Eq.
(A1):
$$SD_1 = 50mm \tag{A1}$$
Soil depth layer 2 (middle) $SD_2$ depends on the absolute depth to bedrock $adb$ – if it is equal or less than 300 mm
computation follow Eq. (A2), otherwise it is conditional of the root depths as per Eq. (A3), and must meet
requirement from Eq. (A4):
$$SD_2 = (adb - SD_1)/2, \, adb \leq 300mm \tag{A2}$$
$$SD_2 = min(root\_depth, (adb - 300mm - SD_1)), \, adb > 300m \tag{A3}$$
$$SD_2 = 50mm, \, SD2 < 50mm \tag{A4}$$
Soil depth layer 3 (bottom) $SD_3$, is computed following Eq. (A5):
$$SD3 = adb - (SD_1 + SD_2) \tag{A5}$$
This set of equations is used twice, once with the root depth of forest area and a second time with the root depth
of non-forested areas, resulting in a total of six soil depth layers computed at SoilGrids250m native resolution.
Soil hydraulic parameters
Soil hydraulic parameters are derived by following three main steps (see Figure A1).
First, soil hydraulic properties are derived at native resolution by applying pedotransfer functions (PTFs) to each
SoilGrids250m soil characteristics layer at each available depth. Pedotransfer functions translate field measured
soil information (such as soil texture, pH and structure) into proprieties and parameters needed to describe soil
processes. The PTFs implemented here are the ones proposed by Toth et al. (2015). Users can decide to derive
soil proprieties from different PTFs, but the general principle presented here remains valid.
Second, the soil hydraulic parameters calculated at SoilGrids250m depths are vertically downscaled to the model
soil depth (previously computed) by weighted average (Figure A1, Step 2 with theta saturated as an example) at
the native SoilGrids250m resolution (~250 m).
Third, the soil hydraulic parameters at the final soil depths are upscaled from native to final resolution by average
using forest and non-forest fraction layers as weights (Figure A1, Step 3).

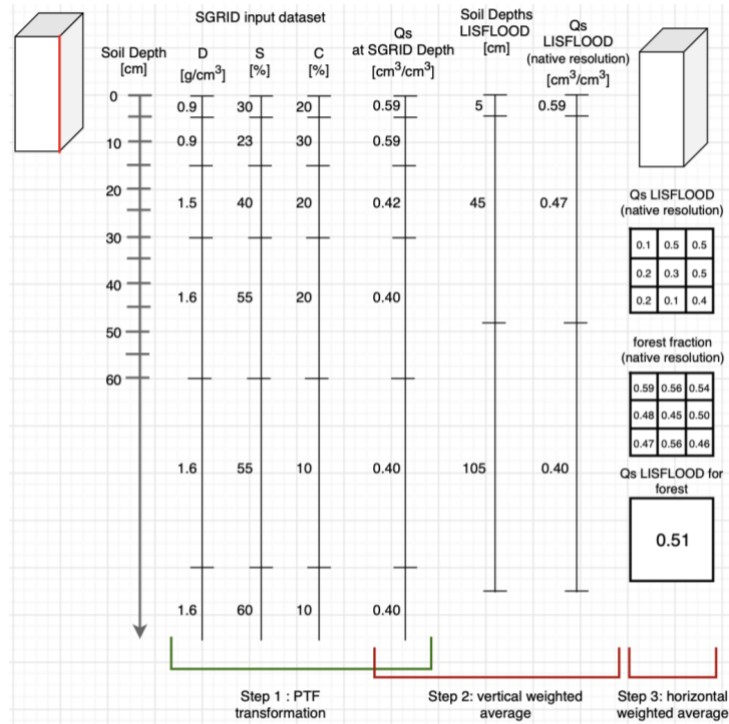

**Figure A1. Creation of theta saturated parameter 'Qs' using SoilGrids250m dataset 'SoilGRID' and forest**
**fraction.**
**Appendix 4**
Here more regional examples of the most interesting surface fields of CEMS_SurfaceFields_2022 are provided to
show what level of details is available at each resolution and field, and to emphasise consistency through all the
fields that is the most valuable requirement when running any type of surface model.

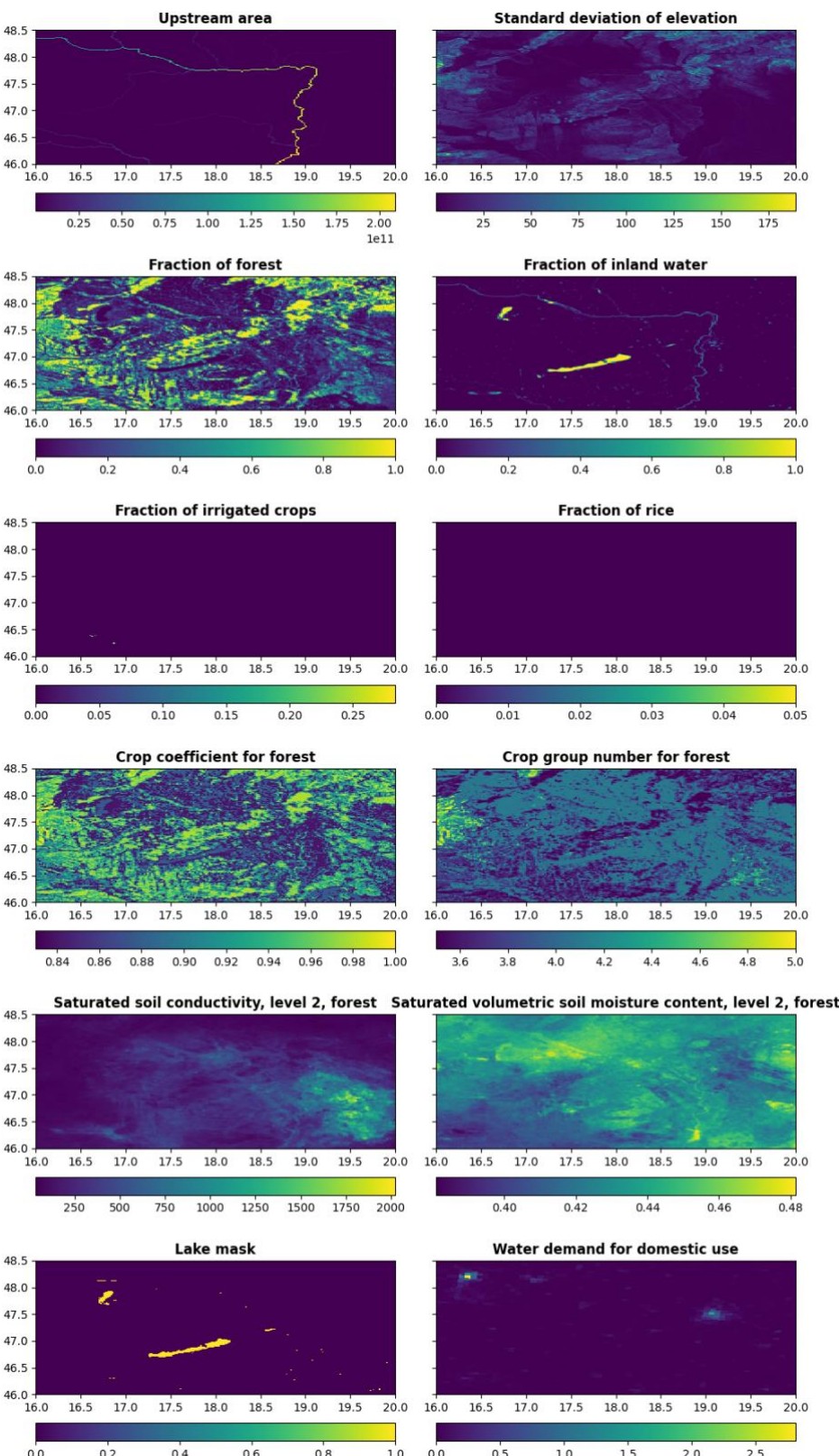

**Figure A2. Upstream drainage area in square meters, standard deviation of elevation in meters, fraction of forest, fraction of inland water, fraction of irrigated crops, fraction of rice, crop coefficient for forest, crop group number for forest, saturated soil hydraulic conductivity for forested areas of soil depth layer 2 in mm per day, saturated volumetric soil moisture (i.e. water) content for forested areas of soil depth layer 2, lake mask, and water demand for domestic use at 1 arc min (~1.9 km at the equator) resolution for Danube River area in Europe.**

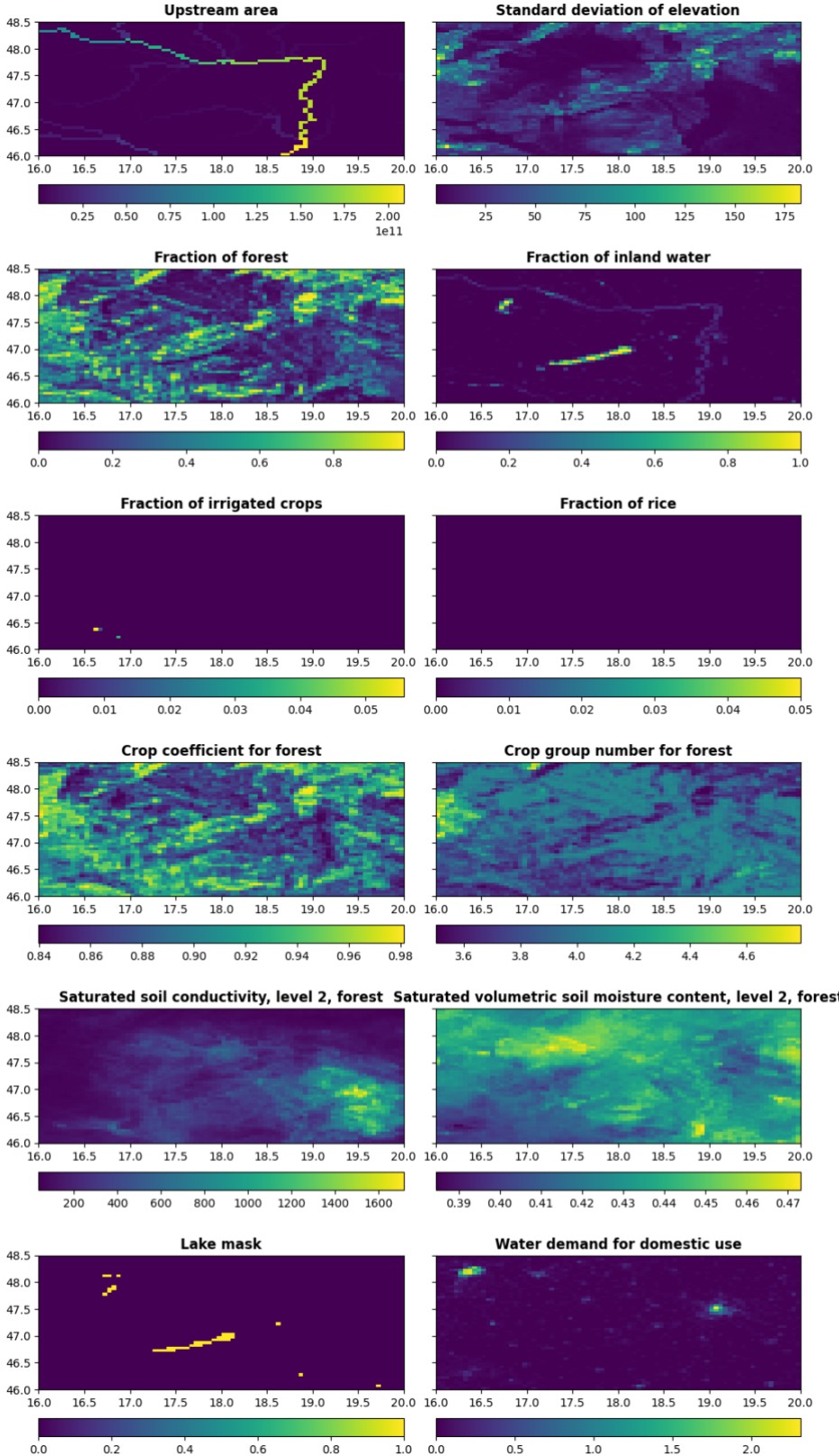

**Figure A3. Same as Figure A2, but at 3 arc min (~5.6 km at the equator) resolution for Danube River area in Europe.**

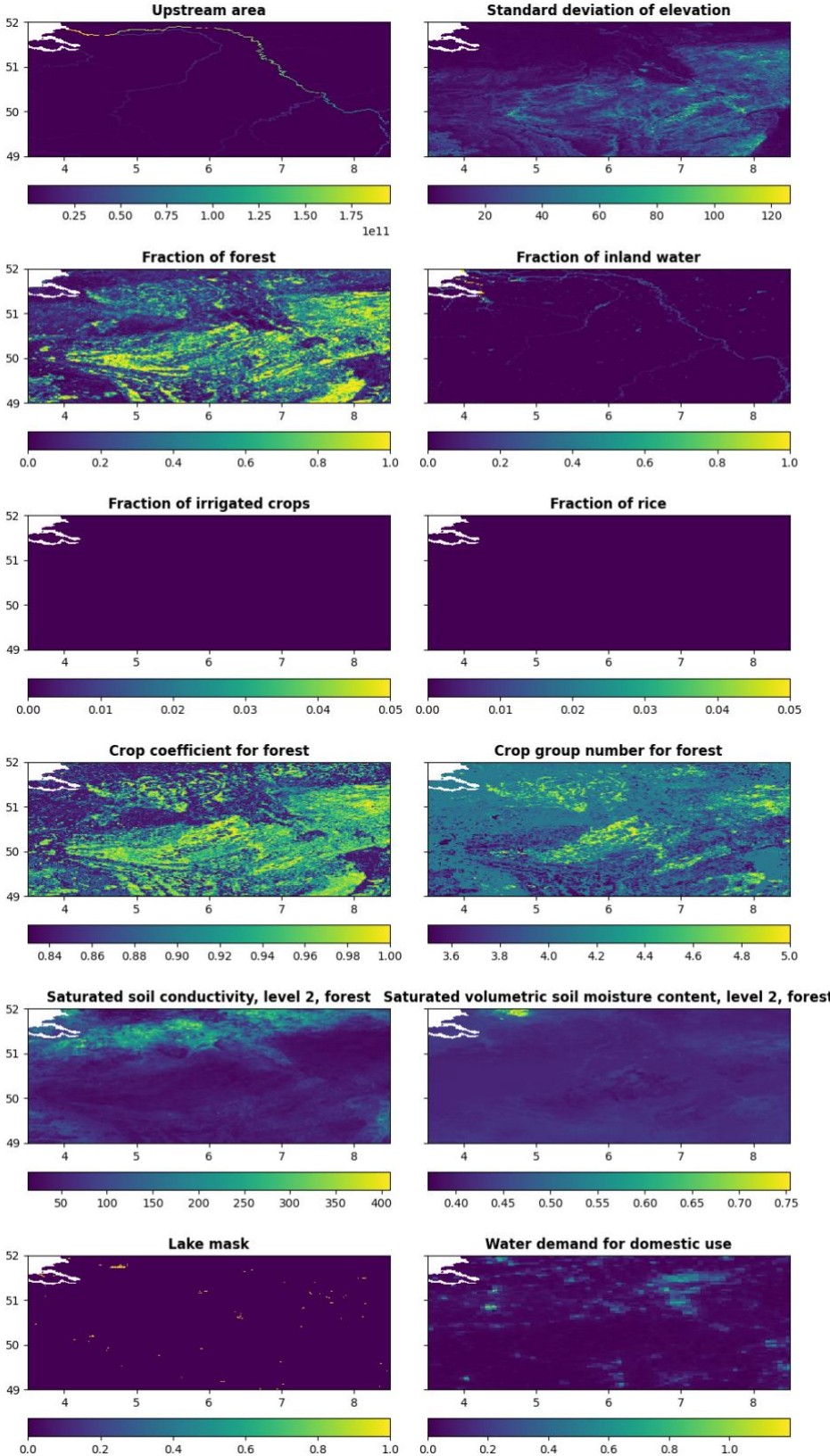

**Figure A4. Same as Figure A2, but at 1 arc min (~1.9 km at the equator) resolution for Rhine River area in Germany.**

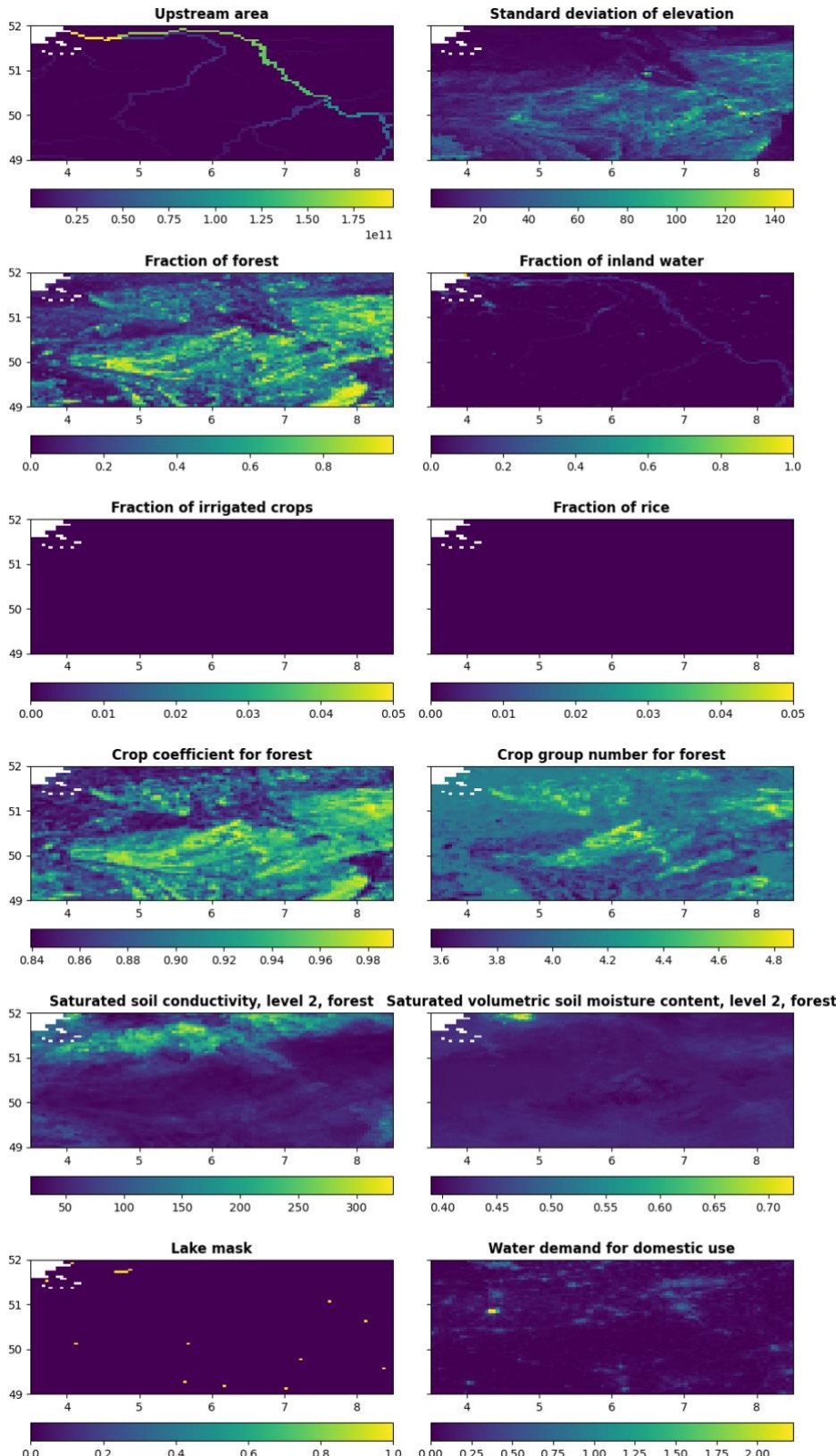

**Figure A5. Same as Figure A2, but at 3 arc min (~5.6 km at the equator) resolution for Rhine River area in Germany.**

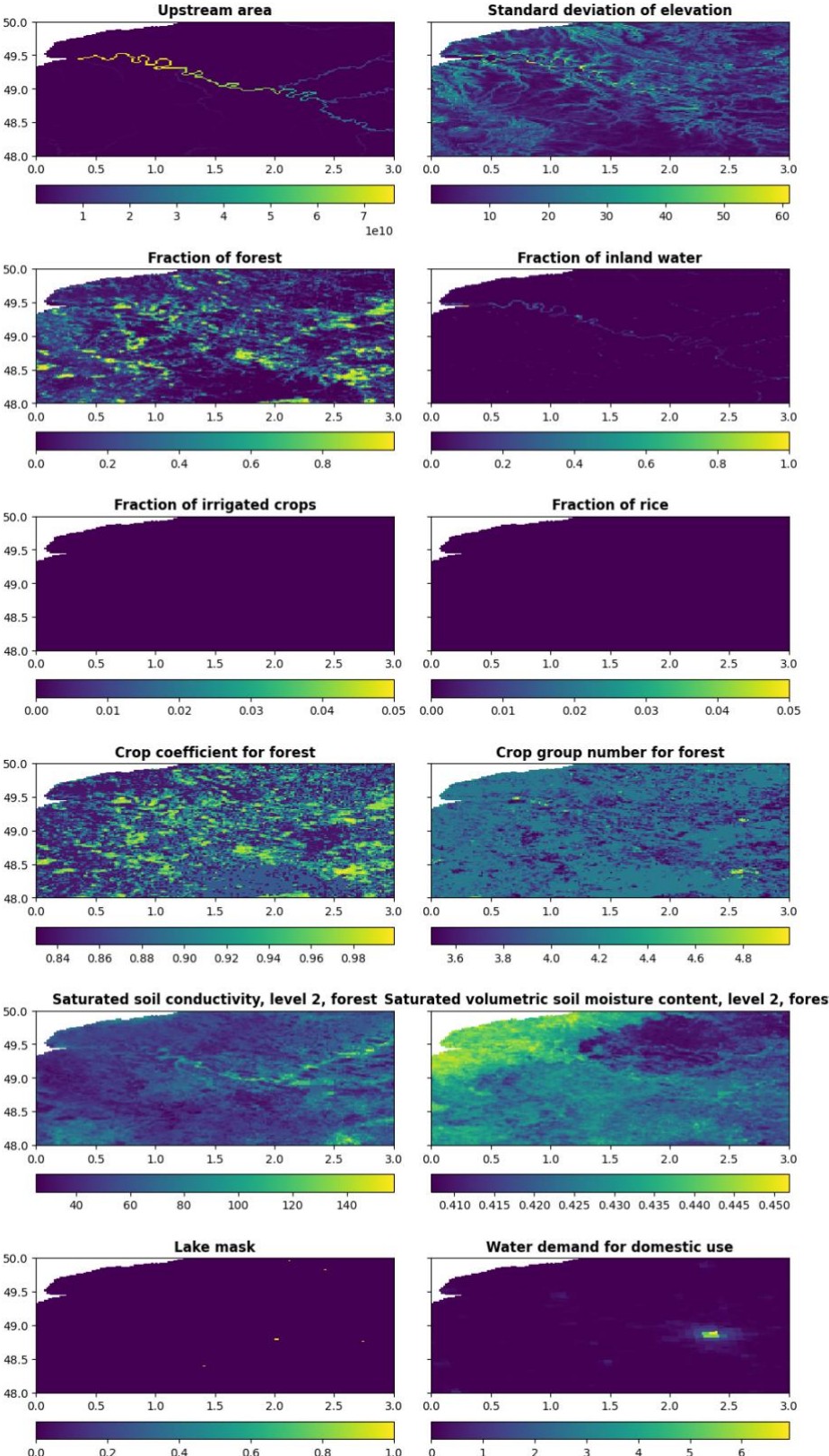

**Figure A6. Same as Figure A2, but at 1 arc min (~1.9 km at the equator) resolution for Seine River area in France.**

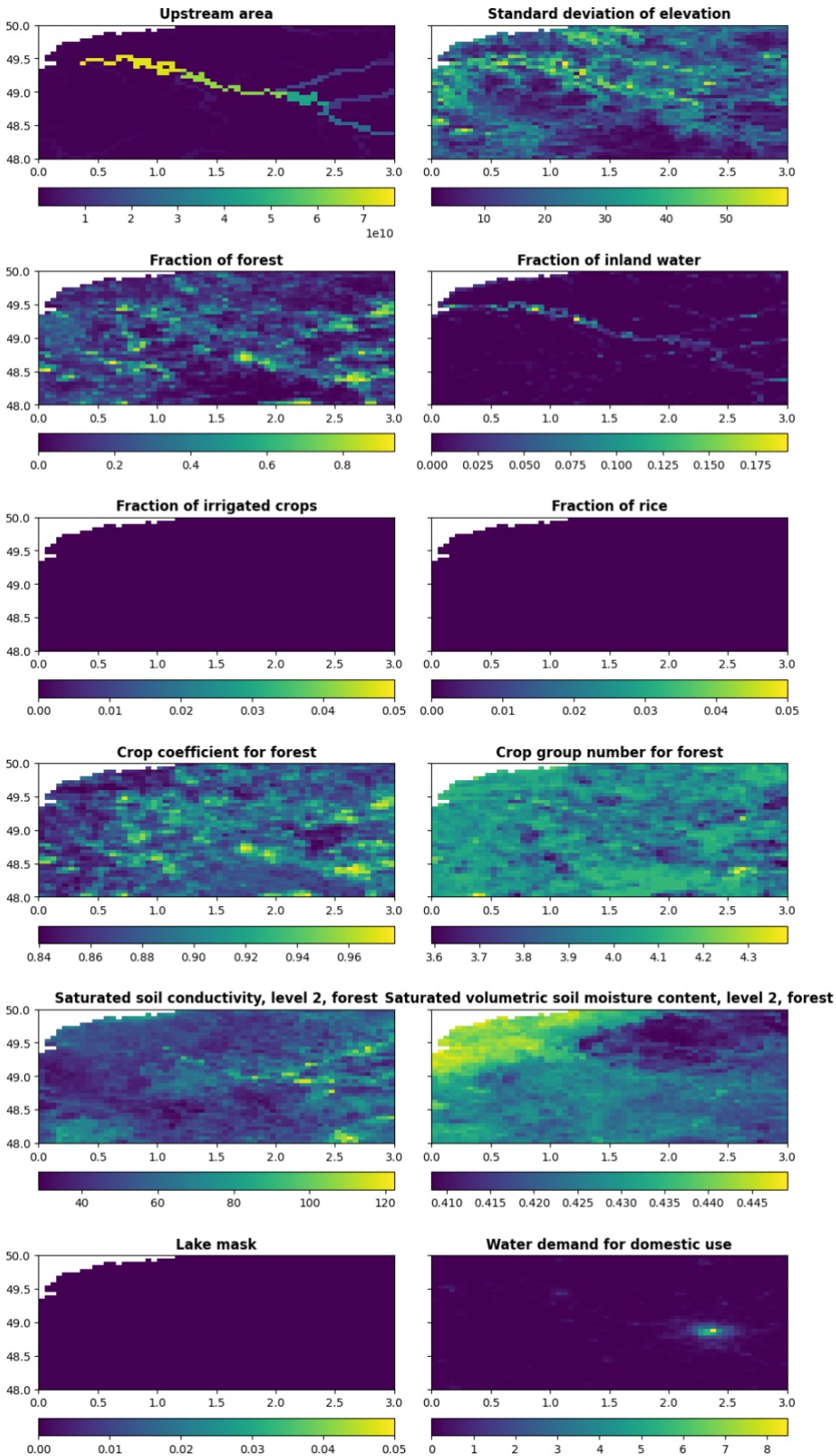

**Figure A7. Same as Figure A2, but at 3 arc min (~5.6 km at the equator) resolution for Seine River area in France.**

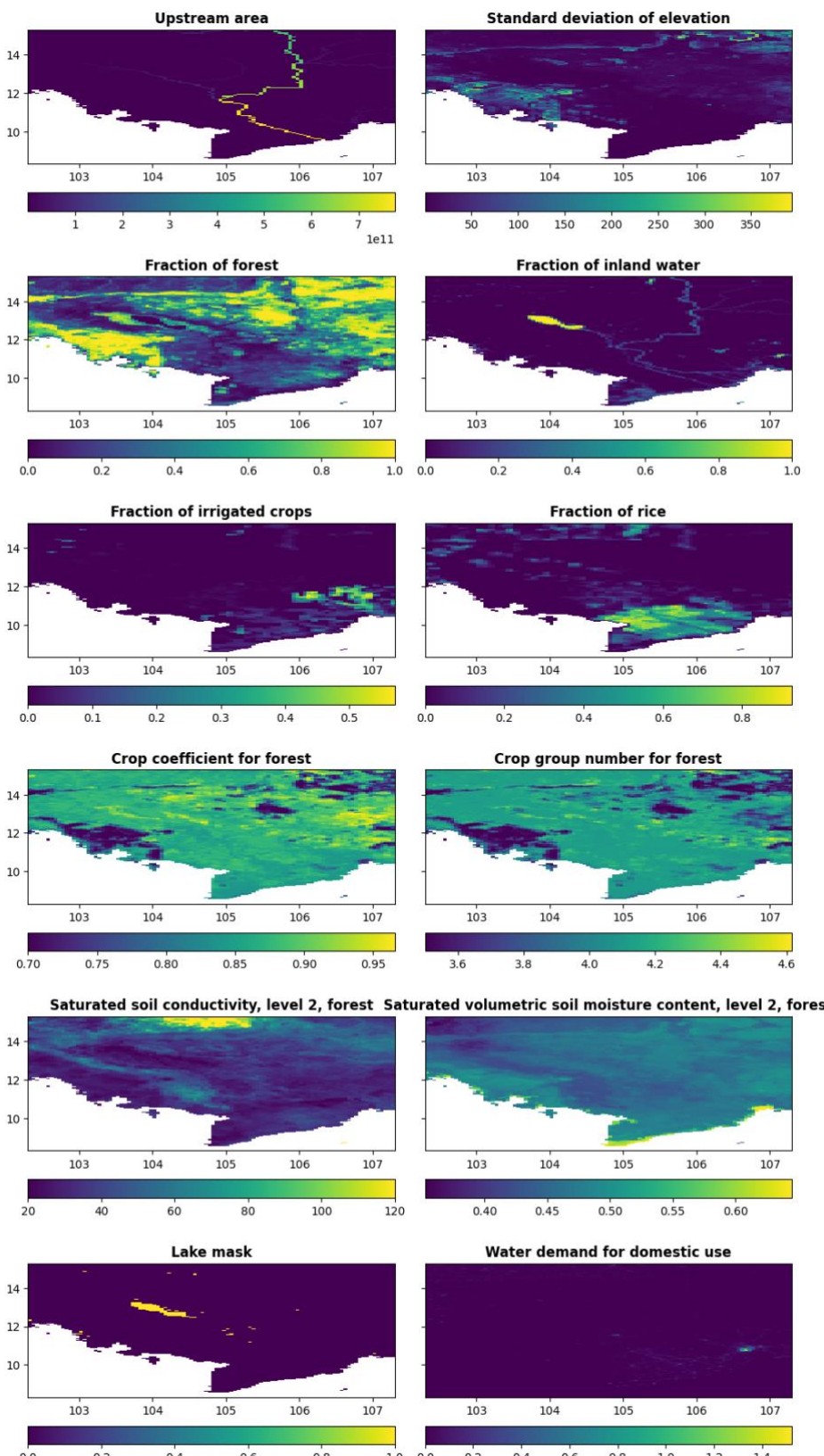

**Figure A8. Same as Figure A2, but at 3 arc min (~5.6 km at the equator) resolution for Seine Mekong area in Cambodia.**