# Peer review of "Technical note: Surface fields for global environmental modelling 1"

_EGUsphere, 2023_

## Author Comment (AC1)

**Responds to Anonymous Referee #1 comments for**
**"Technical note: Surface fields for global environmental modelling" by Choulga et al.**

Dear Anonymous Referee #1,

*General comments*
The authors reported the development of a set of boundary conditions for the global LISFLOOD model at the 3 arc minutes of spatial resolution. The authors' tremendous efforts to develop such high-resolution global data should be well praised. The manuscript is well-written, and I don't have much specific comments.
**Thank you for the positive evaluation.**

My only concern is whether the manuscript fits the technical note of a hydrology-specific journal (HESS). My honest impression is that the manuscript is too specific and looks like a manual of a model. I never intend to depreciate the value of technical documents, but such documents are generally too hard to read for people who haven't been deeply involved in that specific model. My assessments of individual chapters are detailed as follows. Chapter 2 explains the boundary condition of the LISFLOOD model, which is highly model-specific. Chapter 3 lists the datasets chosen by the authors. Many of them are well-known and widely used. It provides neither a general review of available datasets nor the know-how of dataset selection. Chapter 4 explains how the chosen datasets have been converted into the input-ready format of LISFLOOD. Unfortunately, this part is hard to read because most procedure descriptions are squeezed into tables. The text might be informative for LISFLOOD users but too concise to read for non-users. Chapter 5 is short, just specifying metadata. Hence, I skip commenting. Chapter 6 discusses the importance of boundary condition data preparation, but it is not mainly on the data preparation itself but largely refers to the entire operation of the LISFLOOD simulation system. My naïve suggestion is that the manuscript better fits with more data or protocol-oriented papers (e.g. ESSD or GMD). Ideally, it is better to be published as a freely available manual or document of LISFLOOD. If the authors wish this paper to be published in HESS, I recommend further emphasizing the general and applicable wisdom to generate a high-resolution global dataset. Perhaps my assessment above might be useful for revision.
**We agree that the manuscript was compact and contained some repetitions, with the structure that could be improved as also suggested by Reviewer 2. We have totally re-organised the sections, now organised by surface field category, and have moved the description of the source dataset description to an appendix. Finally, for each surface field category we have highlighted possible uses beyond hydrological modelling and provided regional examples of the CEMS_SurfaceFields_2022 dataset.**

*Specific comments*
- Line 105 "The main model's technical field is 'mask'": This part is a bit difficult to read. What do you mean by "technical filed"? **Here we meant that this field is used only by LISFLOOD model, and has little interest for other applications. The sentence was rephrased.**
- Line 514 "Surface field creation overview": I think this subsection should be moved to the next chapter. **We have fully rearranged the manuscript as suggested also by the other reviewer.**
- Line 562: The figure number is missing. **Corrected.**
- Line 570 Table 3: I believe that the authors are mainly explaining the procedures (the row of transformation). Unfortunately, it is hard to read because the sentences are incomplete. Also, some fields seem totally irrelevant to non-LISFLOOD -users (e.g., standard deviation of elevation). **We have rearranged the manuscript and added examples of alternative use for each field category.**
- Line 721 Conclusions: The part is interesting, but it discusses, for example, "simulating long periods (Line 738)", "better modeling (Line 752)," and others. The description of the model and simulation is indispensable in interpreting this part. **We have added information on alternative use and enhanced some of the descriptions.**
- Lines 786- 793: Is this part really needed? **Adapted and rephrased.**

---

## Author Comment (AC2)

**Responds to Anonymous Referee #2 comments for
"Technical note: Surface fields for global environmental modelling" by Choulga et al.**

**Dear Anonymous Referee #2,**

This manuscript describes datasets of surface fields for hydrological modeling, at two resolutions (1 arc min over Europe and 3 arc min over the full globe), for flood awareness system within the Copernicus Emergency Management Service. It merges multiple datasets (more than 25), global and regional, at different spatial resolutions, and with different temporal sampling (from static to 10-day climatology). The consistency of the many data sources is checked, to provide the user with a comprehensive and practical surface dataset for modeling purposes. The dataset is first designed for use with the LISFLOOD hydrological model but is generic enough for adaptation to other models.

*General comments:*
Merging this large diversity of sources is a very valuable effort, and the resulting dataset is very likely to be used broadly, for different modeling purposes. However, this technical note is rather difficult to read, with at the same times too much details and not enough convincing proof that it is a consistent and robust dataset. **Thank you for this feedback. We have restructured the paper and moved details on data source to the appendix, and have added regional high resolution examples to show consistency and robustness of the dataset and examples of applications for each surface field category.**

Sections 2, 3, and 4 repeat the same sequence of parameters (hydrology, vegetation…), first to provide a definition, then to describe the data sources, and finally to present the resulting data in the final dataset. Would it be possible to structure the paper differently and take each parameter at a time, covering first its description, the related data sources and finally how it is handled in the dataset? That would avoid repetitions and could be easier to follow. **Thank you for this suggestion. We have rearranged the sections as suggested and we believe we have now clear and easy to read article.**

The methodology to merge the datasets should be described better, with examples of resulting maps (at high spatial resolution). The reader needs some examples of the consistency of the dataset. A few maps, zooming on specific regions would really help realize what this dataset offers. The global maps that are shown do not present any interesting features, beside what we already know. **To help the reader follow the complex workflow, we have rearranged the text, added flowcharts to each category of surface maps as a guide to the explanation given in the text, and we have added regional examples of the surface field maps.**

*Minor comments:*
- The title should be more specific: Surface 'hydrological' field for global environmental modeling? **We have now added examples of use of the datasets beyond hydrological modelling for each of the surface field category, and we do not feel the title needs changing.**
- The description of the different datasets in section 3 is really difficult to read. There is too much details about most of the datasets. Just provide the reference, the content and general characteristics. **Thank you for this suggestion. We have moved the description of the selected source to the appendix.**
- Why is 'River hydrolics properties' (3.3.2) in the vegetation section (3.3)? **The section was renamed "Vegetation properties" to better reflect its content – surface fields describing how water can move over different land surfaces, e.g. during flooding.**
- Many datasets are merged to produce the water demand information. Any possibility to show that they are collectively consistent? Maps of specific regions? **Maps of specific regions have been added.**
- Figure 4. The sum of the two figures should be 1, no? If this is the case, no use to have both maps. We do not learn much from these large-scale maps anyway (same for figure 5). **Figure 4 shows only two different fractions out of seven that should sum up to 1 per grid cell. We have added regional maps in the main text and in appendix.**
- Line 606: 'Table 5'. **Corrected.**
- Line 696: 'consists' **Corrected.**

---

## Author Response (AR3)

**Responds to Anonymous Referees and Editor comments for**
**"Technical note: Surface fields for global environmental modelling" by Choulga et al.**

Dear Editor,
Thank you for accepting our manuscript for publication.
Since we first submitted initial version of the manuscript, we have totally re-organised it, have added regional examples of the surface fields, have removed any description of the reference data from the main text, have shortened long sentences or cut them in two-three shorter ones, have tried to make text easier to read, and adjusted text where necessary based on two anonymous referees' and your own comments.
Please, find below responds to all the comments.
Sincerely, Margarita

**Responds to Anonymous Referee #1 comments for**
**"Technical note: Surface fields for global environmental modelling" by Choulga et al.**

Dear Anonymous Referee #1,

*General comments*
The authors reported the development of a set of boundary conditions for the global LISFLOOD model at the 3 arc minutes of spatial resolution. The authors' tremendous efforts to develop such high-resolution global data should be well praised. The manuscript is well-written, and I don't have much specific comments.
**Thank you for the positive evaluation.**

My only concern is whether the manuscript fits the technical note of a hydrology-specific journal (HESS). My honest impression is that the manuscript is too specific and looks like a manual of a model. I never intend to depreciate the value of technical documents, but such documents are generally too hard to read for people who haven't been deeply involved in that specific model. My assessments of individual chapters are detailed as follows. Chapter 2 explains the boundary condition of the LISFLOOD model, which is highly model-specific. Chapter 3 lists the datasets chosen by the authors. Many of them are well-known and widely used. It provides neither a general review of available datasets nor the know-how of dataset selection. Chapter 4 explains how the chosen datasets have been converted into the input-ready format of LISFLOOD. Unfortunately, this part is hard to read because most procedure descriptions are squeezed into tables. The text might be informative for LISFLOOD users but too concise to read for non-users. Chapter 5 is short, just specifying metadata. Hence, I skip commenting. Chapter 6 discusses the importance of boundary condition data preparation, but it is not mainly on the data preparation itself but largely refers to the entire operation of the LISFLOOD simulation system. My naïve suggestion is that the manuscript better fits with more data or protocol-oriented papers (e.g. ESSD or GMD). Ideally, it is better to be published as a freely available manual or document of LISFLOOD. If the authors wish this paper to be published in HESS, I recommend further emphasizing the general and applicable wisdom to generate a high-resolution global dataset. Perhaps my assessment above might be useful for revision.
**We agree that the manuscript was compact and contained some repetitions, with the structure that could be improved as also suggested by Reviewer 2. We have totally re-organised the sections, now organised by surface field category, and have moved the description of the source dataset description to an appendix. Finally, for each surface field category we have highlighted possible uses beyond hydrological modelling and provided regional examples of the CEMS_SurfaceFields_2022 dataset.**

*Specific comments*
- Line 105 "The main model's technical field is 'mask'": This part is a bit difficult to read. What do you mean by "technical filed"? **Here we meant that this field is used only by LISFLOOD model, and has little interest for other applications. The sentence was rephrased.**
- Line 514 "Surface field creation overview": I think this subsection should be moved to the next chapter. **We have fully rearranged the manuscript as suggested also by the other reviewer.**
- Line 562: The figure number is missing. **Corrected.**
- Line 570 Table 3: I believe that the authors are mainly explaining the procedures (the row of transformation). Unfortunately, it is hard to read because the sentences are incomplete. Also, some fields seem totally irrelevant to non-LISFLOOD -users (e.g., standard deviation of elevation). **We have rearranged the manuscript and added examples of alternative use for each field category.**
- Line 721 Conclusions: The part is interesting, but it discusses, for example, "simulating long periods (Line 738)", "better modeling (Line 752)," and others. The description of the model and simulation is indispensable in interpreting this part. **We have added information on alternative use and enhanced some of the descriptions.**

- Lines 786- 793: Is this part really needed? **Adapted and rephrased.**

**Responds to Anonymous Referee #2 comments for**
**"Technical note: Surface fields for global environmental modelling" by Choulga et al.**

**Dear Anonymous Referee #2,**

This manuscript describes datasets of surface fields for hydrological modeling, at two resolutions (1 arc min over Europe and 3 arc min over the full globe), for flood awareness system within the Copernicus Emergency Management Service. It merges multiple datasets (more than 25), global and regional, at different spatial resolutions, and with different temporal sampling (from static to 10-day climatology). The consistency of the many data sources is checked, to provide the user with a comprehensive and practical surface dataset for modeling purposes. The dataset is first designed for use with the LISFLOOD hydrological model but is generic enough for adaptation to other models.

*General comments:*
Merging this large diversity of sources is a very valuable effort, and the resulting dataset is very likely to be used broadly, for different modeling purposes. However, this technical note is rather difficult to read, with at the same times too much details and not enough convincing proof that it is a consistent and robust dataset. **Thank you for this feedback. We have restructured the paper and moved details on data source to the appendix, and have added regional high resolution examples to show consistency and robustness of the dataset and examples of applications for each surface field category.**

Sections 2, 3, and 4 repeat the same sequence of parameters (hydrology, vegetation…), first to provide a definition, then to describe the data sources, and finally to present the resulting data in the final dataset. Would it be possible to structure the paper differently and take each parameter at a time, covering first its description, the related data sources and finally how it is handled in the dataset? That would avoid repetitions and could be easier to follow. **Thank you for this suggestion. We have rearranged the sections as suggested and we believe we have now clear and easy to read article.**

The methodology to merge the datasets should be described better, with examples of resulting maps (at high spatial resolution). The reader needs some examples of the consistency of the dataset. A few maps, zooming on specific regions would really help realize what this dataset offers. The global maps that are shown do not present any interesting features, beside what we already know. **To help the reader follow the complex workflow, we have rearranged the text, added flowcharts to each category of surface maps as a guide to the explanation given in the text, and we have added regional examples of the surface field maps.**

*Minor comments:*
- The title should be more specific: Surface 'hydrological' field for global environmental modeling? **We have now added examples of use of the datasets beyond hydrological modelling for each of the surface field category, and we do not feel the title needs changing.**
- The description of the different datasets in section 3 is really difficult to read. There is too much details about most of the datasets. Just provide the reference, the content and general characteristics. **Thank you for this suggestion. We have moved the description of the selected source to the appendix.**
- Why is 'River hydrolics properties' (3.3.2) in the vegetation section (3.3)? **The section was renamed "Vegetation properties" to better reflect its content – surface fields describing how water can move over different land surfaces, e.g. during flooding.**
- Many datasets are merged to produce the water demand information. Any possibility to show that they are collectively consistent? Maps of specific regions? **Maps of specific regions have been added.**
- Figure 4. The sum of the two figures should be 1, no? If this is the case, no use to have both maps. We do not learn much from these large-scale maps anyway (same for figure 5). **Figure 4 shows only two different fractions out of seven that should sum up to 1 per grid cell. We have added regional maps in the main text and in appendix.**
- Line 606: 'Table 5'. **Corrected.**
- Line 696: 'consists' **Corrected.**

**Responds to Editor Frederiek Sperna Weiland decision for**
**"Technical note: Surface fields for global environmental modelling" by Choulga et al.**

**Dear Editor,**

Publish subject to minor revisions (review by editor)

The structure of the Technical Note has been substantially improved - please consider the (minor) review comments and finalize the manuscript for publication.
**Thank you for kindly reviewing our updated manuscript and providing useful comments to improve the manuscript further.**

Additional private note (visible to authors and reviewers only):
This manuscript provides a detailed description of a surface data set that may be of interest to a wide community of modelers. The manuscript has been considerably improved since its first version. Its structure is now clearer, the description of data sources having been partly transferred to an appendix.
**Thank you for acknowledging our hard work.**

It is still a long document, with lengthy descriptions, but as a technical note it can help the data user understand how specific variables have been selected and merged. Nevertheless, efforts still need to be made to make the document shorter and easier to read.
- Some of the information on reference data is duplicated: the information is available not only in the reference data section, but also in the appendix. The reference data section (X.2) should be very short (see for example point 8.2, which is very long). **We have removed any description of the reference data from the main text, only names of the reference data and link to the Appendix 1 are left.**
- Very long sentences and long lists should also be avoided (some are listed below). Many sentences could be cut into several short ones. **We have shortened long sentences or cut them in two-three shorter ones all over the text. We also tried to make text easier to read.**
- The document should be proof-read carefully to remove many typos (see some listed below). **We have re-read the manuscript and traced down some typos.**

Detailed comments:
- Lines 44 to 52: only one sentence…
- Lines 100-103: the sentence is not clear. Rephrase.
- Line 110: is A 'mask'
- Line 113: … model, IT IS a source…
- Line 145: LDD ??
- Line 150: Note THAT Figure 1
- Lines 181 to 184: long sentence. Rephrase.
- Line 188 (and the same sentence at several location): (nameS … correspond…)
- Lines 201 to 209: shorten, as already in appendix… Same for all similar sections.
- Line 219 (and at several occasions in the text): each field correspondS
- Line 242: Application of … includes…
- Line 283: each followS…
- Line 302: other land coverS… - **other land is one of the fraction type, like forest, and inland water; no S was added**
- Line 313-315: With high… missing verb in the sentence.
- Line 338: Alternative use… includeS…
- Line 407: different forest typeS…
- Line 421: soil / water retention
- Line 426: referring TO physical…
- Line 445: an output of special prediction? What do you mean?
- Lines 444 to 454: only one sentence!
- Line 451: 'reference source not found'?
- Line 513: CEMS Surface field: highlighted in green?
- Lines 683 and following. Shorten the conclusion. Why providing details in the conclusion (such as very local remarks at Line 737).
- Lines 757 to 766: only one long sentence!
**All suggestions and comments were implemented by the authors. We really hope that now manuscript can be easier digested by the users and would provide help in understanding surface fields.**